# People infer communicative action through an expectation for efficient communication

Amanda Royka [1 ✉], Annie Chen[2], Rosie Aboody[1], Tomas Huanca[3] & Julian Jara-Ettinger[1,2,4 ✉]

Humans often communicate using body movements like winks, waves, and nods. However, it is unclear how we identify when someone's physical actions are communicative. Given people's propensity to interpret each other's behavior as aimed to produce changes in the world, we hypothesize that people expect communicative actions to efficiently reveal that they lack an external goal. Using computational models of goal inference, we predict that movements that are unlikely to be produced when acting towards the world and, in particular, repetitive ought to be seen as communicative. We find support for our account across a variety of paradigms, including graded acceptability tasks, forced-choice tasks, indirect prompts, and open-ended explanation tasks, in both market-integrated and non-market-integrated communities. Our work shows that the recognition of communicative action is grounded in an inferential process that stems from fundamental computations shared across different forms of action interpretation.

[1] Department of Psychology, Yale University, New Haven, CT, USA. [2] Department of Computer Science, Yale University, New Haven, CT, USA. [3] Centro Boliviano de Desarrollo Socio-Integral, La paz, Bolivia. [4] Wu Tsai Institute, Yale University, New Haven, CT, USA. ✉email: amanda.royka@yale.edu; julian.jara-ettinger@yale.edu

O ther people's behaviors offer a window into their minds: even simple geometrical shapes moving in a two-dimensional space spontaneously elicit the perception of agency and inferences about what the shapes think, want, and feel[1,2]. From early in development, these inferences operate around an assumption that people's actions are typically aimed at interacting with the physical world, like reaching for and manipulating objects (hereafter world-directed goals; refs. [3–9]). By analyzing behavior in terms of world-directed goals, observers can make a wide range of rich social inferences, such as determining other people's goals in physical space, preferences for different objects, competence, knowledge, and even moral standing[10–15].

Yet, many behaviors serve a different purpose: to communicate. When people wave, wink, or nod, their goal is not to act on their physical environment, but to convey a message: acknowledging someone's presence, indicating that they are in on a joke, or agreeing with an argument. How do people recognize that an action is communicative? Past theoretical and empirical work has argued that people solve this problem through a combination of ostensive cues—pre-wired or conventionalized markers of communicative intent (such as eye contact or hearing one's name) and prior experience with the gesture itself[16–21]. Under this view, ostensive cues reveal that someone intends to communicate, and our knowledge of conventional gesture helps us identify which body movements are communicative and which are not (e.g., as someone walks toward you and waves, the person's eye contact reveals communicative intent, and our recognition of the hand-wave reveals that the hand movements are communicative while the leg movements are not). Critically, however, adults can easily identify and respond to new communicative actions without the aid of ostensive cues[22,23] and children can identify, learn, and use gestures effectively before their second birthday[24,25].

We hypothesized that ostensive cues and convention must function in tandem with goal inference: a more flexible, context-sensitive mechanism for identifying that a movement is meant to communicate. Under this view, people do not rely uniquely on cues to recognize communicative action. Instead, they analyze the probable goals behind other people's actions. How would this inferential process affect the recognition of communicative action? Related research has found that people have a strong propensity to analyze movement as world-directed, guided by an assumption that agents move efficiently in space[7,26,27]. This tendency is so strong that, when agents act in seemingly inefficient ways, observers continue to treat the movement as world-directed and invoke more complex explanations, such as inferring additional world-directed goals[10], ignorance[11,12], or situational constraints[13,14]. Here we propose an inferential mechanism to explain how observers infer that an action is communicative: communicative inferences are structured around an expectation that the action will efficiently reveal that it is not world-directed.

What types of movements might efficiently reveal that they are not world-directed? Recent work in cognitive science has developed computational models that infer world-directed goals in a human-like manner via Bayesian inference around an expectation that agents move efficiently in space[10–12,27,28]. We thus used this framework to derive what kinds of movements would, under our hypothesis, lead people to identify communicative action.

Our model-based analysis shows that communicative action should be shaped so that, from the onset, it is unlikely to be produced while pursuing world-directed goals. In other words, communicative movements ought to be *rare* under the distribution of movements that people produce when acting on the physical world (consider, for instance, a body position that is unlikely to ever be generated when interacting with the world, such as a thumbs-up). Although rarity often refers to the statistical frequency of an action,

here we use it as a shorthand to refer to movements that are uncommon when agents pursue world-directed goals. Critically, rarity is related to, but different from inefficiency: some deviations from efficient world-directed action (e.g., deviations due to errors, pursuing subgoals, circumventing hidden obstacles, or movement idiosyncrasies) are inefficient, but still occur when agents pursue world-directed goals[29,30] and are therefore not rare under our definition. Our analysis also revealed one particularly important type of rarity: repetition. That is, agents can make movements rare simply by repeating them, without changing the world. If people expect communicative actions to efficiently reveal that they are not world-directed, then rare and repetitive movements ought to be seen as communicative.

Here we present the results of seven studies that provide evidence for this account. In Studies 1–3, we first test our theory using parametrically varying two-dimensional motions, inspired by those that have been shown to elicit rich mental-state inferences[1,5] and that can be analyzed quantitatively through computational modeling[11,12,14,27]. We find that (i) participants judge rare (and repetitive) paths as more likely to have a communicative goal (Studies 1–2); (ii) that these judgments stem from context-sensitive inference and are not driven by a sensitivity to superficial cues (Study 3); (iii) and that they are driven by intuitions about what communicative action looks like and not by an inability to interpret the movement as world-directed (Explanation Control). Studies 4–7 give further support for our theory using naturalistic videos of unconventional hand movements. Through these videos, we find that people identify rare (and repetitive) hand movements as more likely to be communicative, and that these intuitions can be elicited in a variety of paradigms including direct forced-choice tasks (determining if a movement is communicative), indirect forced-choice tasks (inferring if someone else is in the room, based on how the actor moves), and open-ended explanation tasks (asking people to report what the actor was doing). Moreover, we find the same intuitions among the Tsimane'—a farming-foraging group native to the Bolivian Amazon—providing evidence that these judgments stem from a basic inferential analysis of action, rather than emerging from culturally-dependent pressures.

Throughout, we focused on one-shot events where people must detect conventional communicative action (which was novel to participants) with minimal linguistic or environmental cues. This enabled us to test our hypothesis while controlling for factors that might further affect how people reason about communicative action, including expectations about how form maps to meaning (e.g., iconicity), and how individual gestures fit into complete linguistic systems. We return to these points in the discussion and present the implications of our framework for gestural communication in more complex settings.

## Results

**Studies 1–3**. In Studies 1 and 2, participants watched videos of a point moving in a two-dimensional plane. Participants were told that the point represented a person walking and that their task was to infer whether the movements were produced to communicate with an observer with a bird's eye view (see SI for full cover stories; Fig. 1a). As predicted, we found a strong correlation between rarity (how unlikely it is that the movement would be produced if pursuing a world-directed goal) and communicative inferences (Study 1: $r = 0.80$, $CI_{95\%} = 0.67–1$; replication: $r = 0.87$, $CI_{95\%} = 0.79–1$; Fig. 2a). Moreover, increasing the number of repetitions in the movements while keeping all other features constant led to a significant increase in people's communicativeness ratings (Study 2: $\beta_{\text{repetitions}} = 1.40$, $p < 0.001$; replication: $\beta_{\text{repetitions}} = 1.43$, $p < 0.001$; Fig. 2b).

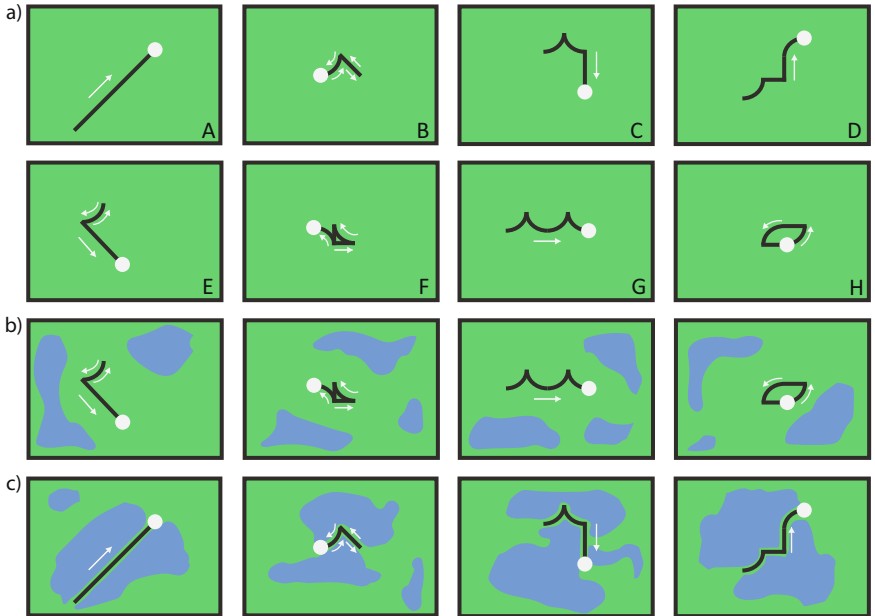

**Fig. 1 Conceptual illustrations of trajectories used in Studies 1–3 and the Explanation Control.** See https://osf.io/ehb48/ for full stimuli set and videos. The trajectories used were a subset of paths built by sequentially chaining combinations of sixteen primitive movements. Paths were sorted into eight movement classes that captured high-level properties of the movements: (A) maximally efficient paths ($n = 2$ paths), (B) paths that retrace themselves back to their origin ($n = 3$), (C) paths that move toward multiple quadrants ($n = 3$), (D) paths that move toward only one quadrant ($n = 3$), (E) paths that partially retrace themselves ($n = 3$), (F) paths that intersect themselves, but do not start and end in the same position ($n = 3$), (G) paths that have repeated components that form a pattern ($n = 3$), and (H) paths that do not retrace themselves, but start and end in the same position ($n = 3$). **a** Example paths used in Study 1. **b** Example paths from the unbordered condition in Study 3. **c** Example paths from the bordered condition in Study 3.

Under our account, the detection of communicative action is inferential and these judgments should therefore be sensitive to the context in which the actions are produced. Specifically, people should cease to treat a movement as communicative when environmental or epistemic factors make a previously rare movement no longer rare (i.e., consistent with a world-directed interpretation). To test this, Study 3 replicated Study 1 (unbordered condition; Fig. 1b) along with a second condition where the same movements could be explained as world-directed due to physical constraints (bordered condition; Fig. 1c). As predicted, participants gave significantly higher communicativeness ratings in the unbordered condition ($M_{Unbordered} = 4.64/7$, $CI_{95\%} = 4.44–4.84$) relative to the bordered condition ($M_{Bordered} = 2.99/7$, $CI_{95\%} = 2.79–3.19$; $\beta_{condition} = 0.87$, $p = 0.002$; replication: $M_{Unbordered} = 4.92/7$, $CI_{95\%} = 4.72–5.12$, $M_{Bordered} = 3.06/7$, $CI_{95\%} = 2.86–3.26$; $\beta_{condition} = 1.23$, $p < 0.001$; Fig. 2c), showing that communicativeness judgments were indeed inferential and not driven by a movement's superficial structure.

Our results so far show that people judge rare and repetitive paths as more likely to be communicative. There are at least two possible mechanisms behind these judgments. A first possibility is that people can flexibly interpret the movement as world-directed or communicative, but find communicative interpretations to be more suitable for rare and repetitive movements (based on the relative likelihood of observing the behavior under each interpretation). A second possibility is that people are simply unable to interpret rare and repetitive movements as world-directed, and therefore default to a communicative interpretation. While both processes are consistent with our account, the second one raises a methodological concern: is it possible that participants did not believe that the movements looked communicative, and their responses reflected only confidence that the movements could not possibly be world-directed?

To explore this possibility, we tested participants' ability to invoke non-communicative explanations for all Study 1 movements. We reasoned that if participants can easily invoke non-communicative explanations, independent of path rarity and repetition, then Study 1 participants were likely actively endorsing a communicative interpretation (rather than defaulting to it because they were at a loss about how else to interpret the movement). Participants in the Explanation Control watched the same videos from Study 1, but were given a context where communication was unlikely (see "Methods"). Participants were asked to explain the agent's goal and rate how difficult it was to conceive of this explanation. If Study 1 judgments reflected an inability to think of non-communicative goals, then path rarity should correlate with difficulty of explanation (revealing that participants rated rare paths as communicative only because it was difficult to think of an alternative interpretation, replicating the correlation found in Study 1). Instead, we found no significant relation between rarity and difficulty of explanation (Explanation Control: $r = 0.10$, $CI_{95\%} = −0.31–0.56$, $\beta_{rarity} = 0.21$, $p = 0.43$; replication: $r = 0.38$, $CI_{95\%} = 0.14–0.83$, $\beta_{rarity} = 0.40$, $p = 0.19$; Fig. 2d), suggesting that people can easily conceive of non-communicative goals for complex movements. Thus, responses in Studies 1–3 likely primarily reflect participants' belief that a communicative goal was a better explanation, rather than defaulting to this answer due to an inability to think of alternatives (see SI for additional analyses revealing this null effect was not due to task misunderstanding, and Studies 6–7 for additional evidence that rare and repetitive movements are actively seen as communicative).

**Modeling communicative inferences.** Our first study approximated rarity by quantifying deviations from the shortest path, but not all inefficiencies are rare (see "Introduction"). To test a more nuanced view of rarity, we took computational models of goal inference[10,11,12,28] and modified them to compute the likelihood that movements might appear communicative. Critically, these models can naturally distinguish between inefficiencies that are consistent with world-directed behavior (e.g., zig-zagging toward

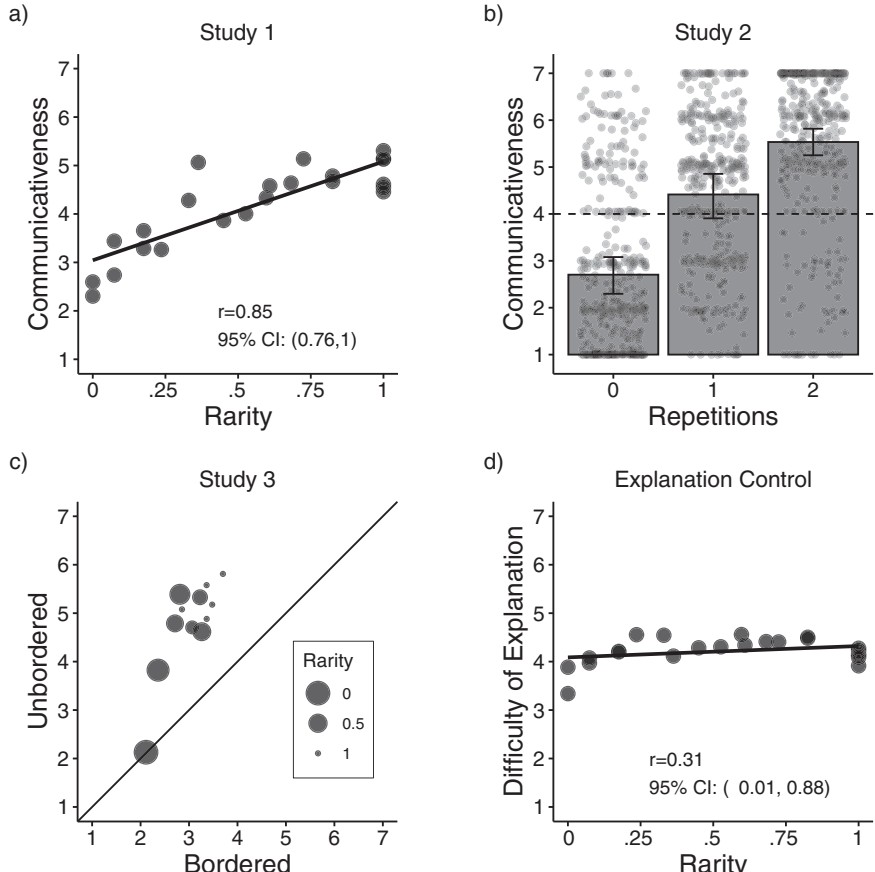

**Fig. 2 Results from Studies 1–3 and the Explanation Control, combining original studies and their pre-registered replications. a** Average communicativeness judgments (y-axis) as a function of the path's rarity (x-axis) in Study 1 ($n = 60$), operationalized here as deviation from the shortest path (see "Methods"). **b** Average communicativeness judgments in Study 2 (y-axis) as a function of number of path repetitions (x-axis; $n = 60$). Vertical lines show bootstrapped 95% confidence intervals. Dots represent individual judgments. **c** Study 3 results as a function of rarity and condition. Each point represents a path with the average communicativeness in the bordered condition (x-axis) and the unbordered condition (y-axis). Smaller circles represent rarer paths. Points above the diagonal line indicate a higher communicativeness rating in the unbordered condition relative to the bordered condition. The difference across conditions was larger for paths with higher rarity in a mixed-effects model ($\beta_{\text{rarity:condition}} = 1.14$, $p = 0.002$, $n = 30$; replication: $\beta_{\text{rarity:condition}} = 0.92$, $p = 0.015$, $n = 30$), further suggesting that these paths were no longer seen as communicative because environmental constraints in the bordered condition removed the paths' perceived rarity. **d** Average reported difficulty of generating a world-directed explanation as a function of the path's rarity (x-axis) in our Explanation Control ($n = 60$). If Study 1 judgments were driven by a pure inability to consider world-directed explanations, then the difficulty of explanation should show a strong positive correlation with rarity, but this was not the case.

a goal) and inefficiencies that cannot be explained by any world-directed goal. In this framework, the probability that an agent is pursuing a world-directed goal in a physical environment $W$ is given by

$$p(W|a) \propto \sum_{g \in G} p(a|g)p(g|W)p(W) \quad (1)$$

where $G$ is the space of possible world-directed goals an agent may pursue, and $a$ are the observed actions (see SI for details). Thus, our model considers the path's likelihood under every possible world-directed goal ($p(a|g)$), weighted by the prior probability that the agent would pursue each world-directed goal ($p(g|W)$) using a uniform distribution (as participants in our task had no information about the probable position of world-directed goals; see SI).

As Fig. 3a shows, paths that the model found to more quickly reveal that they lack a world-directed goal (those with sharper slopes) were the same ones that participants rated as more communicative, revealing a correlation of $r = 0.77$ (CI$_{95\%}$: 0.64–1) between model predictions and participant judgments (Fig. 3b). Critically, however, these predictions only considered the final

belief that a movement was not world-directed, without taking into account how quickly this occurred. Under our proposal, movements that reveal non-world-directedness quickly should be seen as more communicative relative to paths that reveal non-world-directedness slowly. To test this, we analyzed the cases where people gave higher or lower communicativeness judgments relative to our model predictions. Figure 3c shows these results as a function of the eight movement classes used in Study 1 (see Fig. 1a for an example of each movement class and Fig. 1 caption for explanations of classes). The largest discrepancies appeared in two classes: First, the model under-estimated communicativeness in the repetitive paths (class B). This suggests that, in line with our account, paths that quickly reveal their non-world-directedness through repetition are seen as more communicative relative to paths that do so in a more protracted manner. Second, the model over-estimated communicativeness in paths that inefficiently moved toward a single quadrant (class D). This suggests that paths that ultimately reveal that they lack a world-directed goal, but that fail to do so quickly (as the movement is initially consistent with world-directed goals), are seen as less communicative by people relative to the model. Indeed, removing

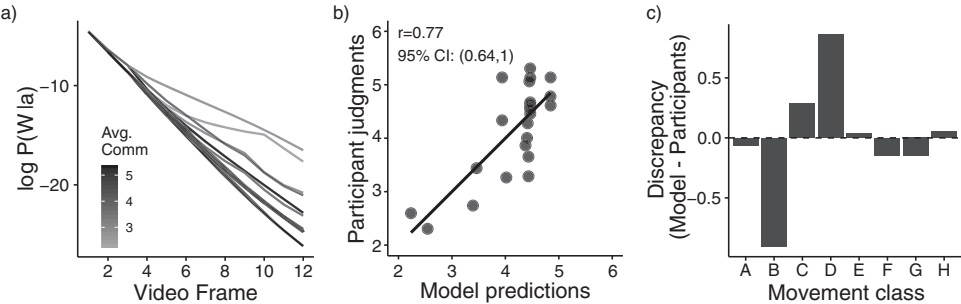

**Fig. 3 Results from a computational model that determines communicativeness when the movement reveals that it lacks a world-directed goal.**
Critically, this model has a more nuanced concept of rarity, as it can infer world-directed goals for inefficient movements (such as an agent zig-zagging toward an object). **a** Model predictions showing the belief that each path from Study 1 was world-directed as a function of time. Each line represents one of the 23 videos from Study 1 (partially occluded due to over-plotting), with frame number on the x-axis and model prediction on the y-axis. Lower model predictions indicate that the movement looked less world-directed (in log-space; see Eq. 1). Each line's shading indicates the average communicativeness rating received for that video in Study 1. **b** Final model predictions (x-axis) against participant judgments (y-axis) from Study 1. To make scales comparable, we transformed model ratings into communicative inferences through a linear regression predicting participant judgments based on the model's final output (see SI for details). Each point represents a path's model prediction (x-axis; negative log of probability of a world-directed goal given the full trajectory—equivalent to the predictions on frame 12 of panel a)—against average communicativeness ratings from Study 1 (y-axis). **c** Disagreement between model predictions and participant judgments as a function of the movement class (see Fig. 1 for movement class labels). Positive numbers indicate that the model saw the movement as more communicative than participants and negative numbers indicate that the model saw the movement as less communicative than participants.

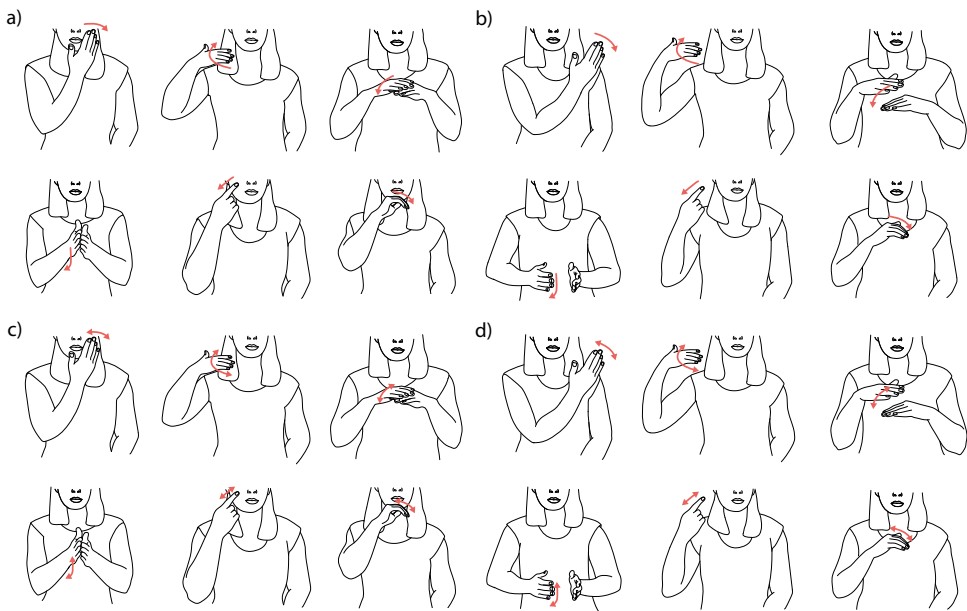

**Fig. 4 Schematic of body movements used in Studies 4–7.** See https://osf.io/wxdka/ for videos. **a–d** *Basic* movements, *rare* movements, *repetitive* movements, and *rare + repetitive* movements. The actor's eyes were never visible and the face was always directed at the camera, preventing participants from relying on facial information. All videos were normed to ensure that they exhibited rarity and repetition. See SI for norming data on videos.

paths from classes B and D significantly increased the correlation between model predictions and communicativeness judgments from $r = 0.77$ to $r = 0.90$ ($p < 0.001$ by permutation test). Together, these results show further evidence that people infer that an action is communicative not only when it reveals the absence of a world-directed goal, but more so when this is efficiently achieved.

**Studies 4–7.** Having found support for our account in controlled displays, we next sought support for our theory in more naturalistic situations. To achieve this, we created short videos of a woman producing different hand movements (Fig. 4). All movements were based on world-directed goals (e.g., wiping off hands, moving hair) from which we created four versions of each

movement: a *basic* version (Fig. 4a), in which the woman moved as if to accomplish a world-directed goal; a *rare* version (Fig. 4b), in which the woman produced similar movements to those from the basic version, but displaced so that they were unlikely to be produced when pursuing any world-directed goal; a *repetitive* version (Fig. 4c), in which the woman repeated the basic movement three times; and a *rare + repetitive* version (Fig. 4d), in which the woman repeated the rare movement three times.

In Study 4, participants were asked to judge their confidence that different movements were communicative. Because brief pauses and sharp changes in speed and direction—or *punctuality*—are typical of representational gestures and thought to be cues to communicative intent[31–33], Study 4a used a demonstrator that limited the punctuality of all movements (*low punctuality* videos).

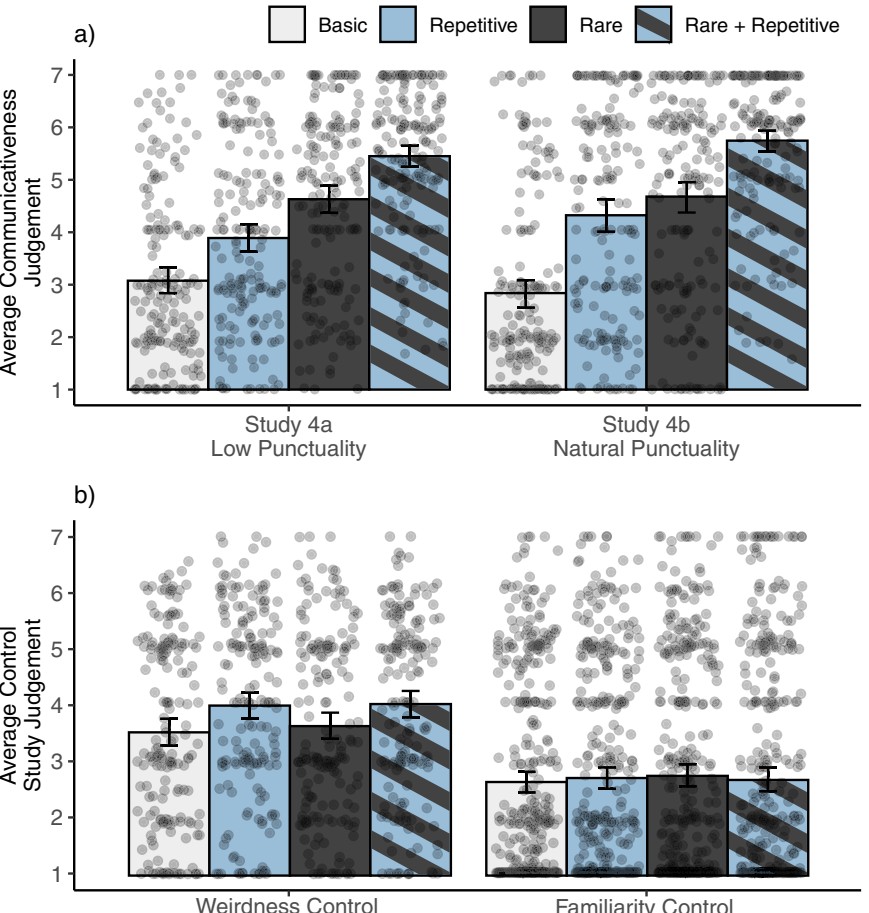

**Fig. 5 Results from Study 4a, Study 4b, and the two controls.** Each bar shows the average judgment and vertical lines show bootstrapped 95% confidence intervals. Dots represent individual judgments. **a** Average communicativeness judgments from US participants for Low Punctuality (Study 4a; $n = 30$) and Natural Punctuality (Study 4b, $n = 30$) demonstrators broken down by video type. **b** Results from the Weirdness Control and Familiarity Control studies. Participants' beliefs about a movement's weirdness (Weirdness Control; $n = 30$) or familiarity (Familiarity Control; $n = 60$) failed to explain our results.

To also ensure that the actor's knowledge of the communicative context of the study and the effects of punctuality did not affect our results, Study 4b replicated Study 4a with videos featuring a naïve demonstrator who was unaware that the study was about communicative action (*natural punctuality* videos). Rarity and repetition significantly predicted communicativeness ratings in both studies (Fig. 5a; $\beta_{rarity} = 1.55$, $p < 0.001$; $\beta_{repetition} = 0.82$, $p < 0.001$ in Study 4a by a mixed-effects regression; $\beta_{rarity} = 1.63$, $p < 0.001$; $\beta_{repetition} = 1.27$, $p < 0.001$ in Study 4b by a mixed-effects regression), and the results were not significantly different as a function of punctuality ($\beta_{punctuality} = -0.14$, $p = 0.49$ in a mixed-effects regression combining the two studies; see SI).

Is it possible that participants determined whether a movement was communicative based on how unusual it looked? To test this possibility, participants in the Weirdness Control were asked to rate how weird each movement from Study 4a appeared. Judgments in the Weirdness Control did not replicate the qualitative pattern from Study 4 (Fig. 5b), and the effects of rarity and repetition were significantly stronger in our test condition relative to the control ($\beta_{question:rarity} = 1.48$, $p < 0.001$ and $\beta_{question:repetition} = 0.39$, $p = 0.007$). A related possibility is that movements rated as communicative were inadvertently similar to or reminiscent of existing communicative gestures. To test this possibility, participants in the Familiarity Control watched the same videos from Study 4, but were now told that all movements were gestures and

they were asked to rate how familiar each gesture appeared. Familiarity ratings were low across all video types ($M_{Basic} = 2.67$, $M_{Rare} = 2.78$, $M_{Repetitive} = 2.74$, $M_{Rare+Repetitive} = 2.71$) and significantly lower relative to communicativeness judgments ($\beta_{question:rarity} = 1.55$, $p < 0.001$ and $\beta_{question:repetition} = 1.05$, $p < 0.001$), failing to replicate the pattern from Study 4 (Fig. 5b).

According to our proposal, communicative action is detected by relying on basic action-understanding mechanisms that emerge early in infancy[3–6,8], and these inferences might therefore show cross-cultural stability. As a first step to explore this question, we evaluated these intuitions with the Tsimane'—a farming-foraging group living in the Bolivian Amazon. The Tsimane' differ from US participants in several areas including the timing of the acquisition of number words and concepts[34,35], aesthetic preferences in sound[36], and color vocabulary[37]. Studies 5–7 were designed in consultation with our interpreters and local experts, and then replicated in the US. In Study 5, we first presented Tsimane' participants with pairs of videos (a rare or a repetitive video paired with its corresponding basic video; Fig. 4) and asked them to determine which of the two movements was communicative (Study 5). Tsimane' participants were significantly more likely to indicate that the rare and repetitive movements were communicative in both the low punctuality videos (Fig. 6; rare: 56.11% of trials, $CI_{95\%} = 50.28–61.94\%$, $p = 0.002$ by permutation test; repetitive: 63.06%,

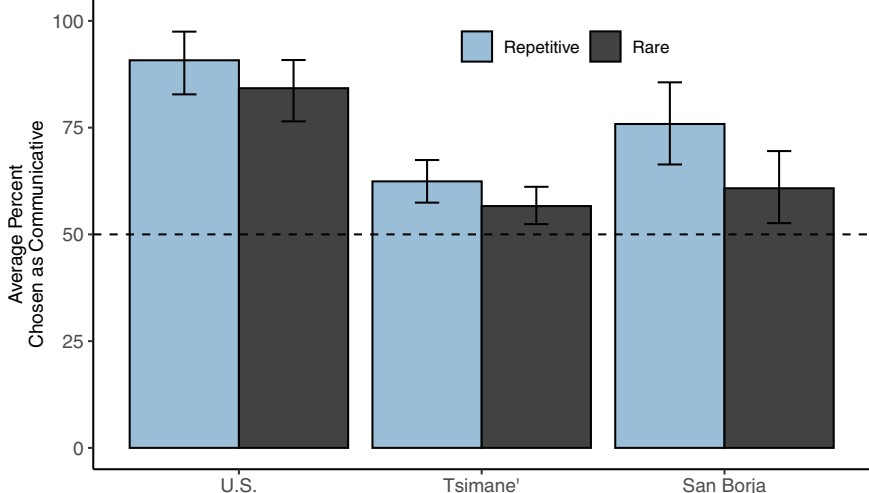

**Fig. 6 Results from Study 5.** Left to right: Average preference for the rare or repetitive video as communicative with US, Tsimane', and San Borja participants. San Borja is the Spanish-speaking market-integrated town that is closest to the Tsimane' communities, which served as a geographic control. Each bar shows the average percentage of endorsements for rare or repetitive movement when directly contrasted with a basic one. U.S. ($n = 40$) and San Borja ($n = 40$) participants completed the task with the low punctuality demonstrator videos. Tsimane' participant results are combined across both low punctuality ($n = 120$) and natural punctuality videos ($n = 60$). Vertical lines show bootstrapped 95% confidence intervals.

$CI_{95\%} = 56.67$–$69.17\%$, $p < 0.001$) and the natural punctuality videos (rare: 57.78%, $CI_{95\%} = 50.56$–$64.44\%$, $p = 0.002$; repetition: 61.11%, $CI_{95\%} = 52.78$–$69.44\%$, $p = 0.001$).

Furthermore, we ran the same forced-choice paradigm with US participants and with San Borjans—residents of the market-integrated town in the Bolivian Amazon that is geographically closest to the Tsimane' communities (Study 5). Like the Tsimane', both San Borjan and US participants were significantly more likely to indicate that the rare and repetitive videos were communicative (San Borjan, rare: 60.83%, $CI_{95\%} = 52.50$–$69.17\%$, $p < 0.001$; repetition: 75.63%, $CI_{95\%} = 65.62$–$84.79\%$, $p < 0.001$; US, rare: 84.17%, $CI_{95\%} = 76.67$–$90.83\%$, $p < 0.001$; repetitive: 90.83%, $CI_{95\%} = 82.5$–$97.5\%$, $p < 0.001$) relative to the basic movements. This graded increase in performance, rather than a sharp discontinuity between non-market-integrated communities (Tsimane') and market-integrated communities (San Borjans, and US participants), may suggest a universal tendency to infer communicative goals for movements that efficiently reveal they lack a world-directed goal, with the strength of this effect varying as a function of familiarity with experimental procedures (see ref. [38] for a parallel effect of formal schooling on task performance). However, it is also possible that the difference in effect sizes emerged because participants from different cultures interpreted the task in different ways (known as a lack of *equivalence*; refs. [39–43]), and our results should be interpreted with caution. As such, these results represent a first step in establishing that Tsimane' and San Borjan participants are also more likely to endorse rare and repetitive movements as potentially communicative.

While consistent with our account, Study 5 used a forced-choice paradigm and we did not know the extent to which participants were endorsing the rare and repetitive videos, or rejecting the basic ones. We therefore next tested whether this effect appeared when using indirect questions and open-ended explanation paradigms. In Study 6, US and Tsimane' participants were told that they would watch short videos of a single person (Fig. 4a, d) and that their task was to infer if there was a second person in the room. If participants viewed the rare + repetitive movements as communicative, we expected them to infer that there must be another person whom the gesture was directed at. As predicted, both US and Tsimane' participants inferred the presence of another person in the room when watching a

rare + repetitive video (Tsimane': 78.53% of trials in which participants responded that there was another person, $CI_{95\%} = 71.19$–$85.31\%$, $p < 0.001$; US: 94.92%, $CI_{95\%} = 91.53$–$97.74$, $p < 0.001$). By contrast, participants were significantly more likely to report that the protagonist was alone when watching a basic video (Tsimane': 67.80% of trials, $CI_{95\%} = 61.02$–$74.58\%$, $p < 0.001$; US: 74.58%, $CI_{95\%} = 67.23$–$81.36\%$, $p < 0.001$; see Fig. 7a and SI).

Finally, we ran an open-ended explanation task (Study 7), where US and Tsimane' participants were asked to explain what they thought the demonstrator was doing (using basic and rare + repetitive low punctuality movements). Participant responses were then categorized as *world-directed* (aimed at producing a change in the physical world), *communicative* (produced with the goal of sharing a message with an observer), *descriptive* (describing the low-level actions performed by the demonstrator), or *other* (a catch-all category) by two coders who were blind to the video behind each explanation. Among the subset of responses that were categorized as either world-directed or communicative (63% of responses from US participants and 92.7% of responses from Tsimane' participants), communicative explanations were most often produced in response to rare + repetitive videos (US: 80.77% of rare + repetitive video trials, $CI_{95\%} = 74.73$–$86.26\%$; Tsimane': 61.80%, $CI_{95\%} = 51.69$–$71.91\%$; Fig. 7b; see SI for supporting mixed-effects model results) while world-directed explanations were most often produced in response to basic videos (US: 92.86% of basic video trials, $CI_{95\%} = 89.29$–$96.43\%$; Tsimane': 84.27%, $CI_{95\%} = 76.40$–$91.01\%$). Results are qualitatively the same when all responses are included (see SI).

## Discussion

Our work shows that the ability to recognize and interpret body movements as communicative is grounded in a context-sensitive inferential process, where observers expect communicative action to efficiently reveal that it is not meant to change the physical world (i.e., not world-directed). Critically, people's communicative inferences were not determined purely by movements' superficial structure and they were sensitive to whether contextual information supported a world-directed interpretation. These results are consistent with everyday situations where rarity and repetition are often present in communicative action (e.g., waving, nodding, and winking), but are treated as world-directed

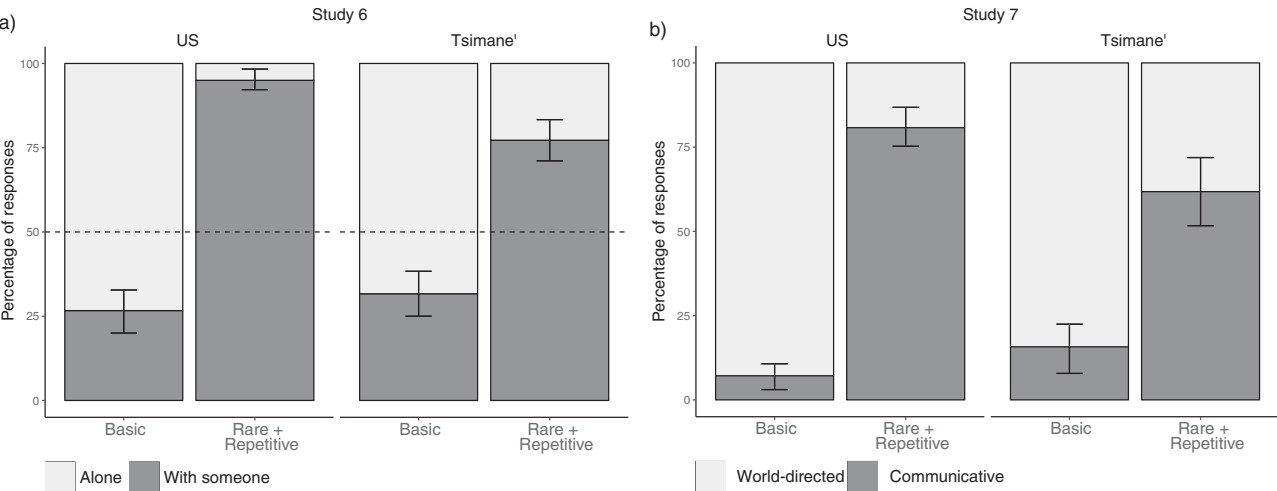

**Fig. 7 Results from Studies 6 and 7.** Vertical lines represent 95% bootstrapped confidence intervals. **a** Average percentage of trials in Study 6 where US ($n = 59$) and Tsimane' ($n = 59$) participants judged that the actor was alone or with someone as a function of the type of video they watched. Participants were more likely to judge that the actor was alone when watching a basic movement, and more likely to infer the presence of another agent when watching a rare + repetitive movement. **b** Average percentage of world-directed and communicative explanations that US ($n = 100$) and Tsimane' ($n = 32$) participants produced as a function of video type (Study 7). Results are normalized after excluding explanations coded as *descriptive* or *other* (see SI). Participants were more likely to produce a world-directed explanation when watching a basic movement, and significantly more likely to produce a communicative one when watching a rare + repetitive one (this result does not depend on the pre-registered exclusion of *descriptive* and *other* explanations; see SI).

when the context supports this interpretation (e.g., scratching is interpreted as world-directed despite its repetition). Indeed, work in ethnography has noted that people often hide offensive gestures by producing them in a context that supports a world-directed interpretation, giving them plausible deniability (e.g., someone using a middle finger to scratch their eye; refs. [44–46]).

Altogether, our studies suggest that rarity and repetition support communicative inferences because they efficiently reveal non-world-directedness in a wide range of contexts. In the remainder of the discussion, we begin by considering two key limitations left unaddressed by our current work and then discuss two open questions for future investigation. Finally, we elaborate on the connections between our framework and related areas of research.

A first limitation of our work is that our studies treated world-directed and communicative goals as mutually exclusive, but this is often not the case. Goals have a fundamental hierarchical structure and they can be explained at varying levels of abstraction. For example, a single action performed in front of an observer may be conceptualized as moving a hammer up and down, hammering a nail, and showing how to properly use a hammer. From this standpoint, our studies not only show that people see rare and repetitive movements as communicative, but that the features that trigger this inference may also lead people to favor this level of explanation over lower-level ones (particularly in our open-ended response task; Study 7). Consistent with this, related work has found that, when agents produce rare or repetitive movements in a context where both communicative and world-directed goals are implausible, people default to the lowest-level explanation, judging that the agent's goal was to simply produce the movements themselves[7]. In these cases, however, a communicative-level explanation is recovered when an observer is present (see SI for an additional study replicating the initial effect and an extension showing when communicative-level explanations are recovered).

A second limitation of our work is that we exclusively focused on rarity (and repetition, as a mechanism for making movements rare) as a way to efficiently reveal that a movement is not world-directed. This category, however, was derived by considering a space of simple goals similar to those that even infants

understand (i.e., agents moving efficiently in space toward target locations; refs. [6,8,26]). It is likely that additional features would emerge under a richer model of world-directed action (particularly models that include the inefficiencies associated with world-directed goals, such as those introduced by incompetence or information seeking). Indeed, a movement's velocity, size, and punctuality also influence the recognition of communicative action[47–50]. These features might further support the recognition of communicative action precisely because they efficiently reveal that the movement is not world-directed. Thus, our work provides broad support for the idea that communicative action is expected to efficiently reveal a lack of world-directed goals, but leaves open the question of how people expect this principle to shape the spatiotemporal structure of action in more complex events.

Our work also opens two central questions regarding the relationship between signaling communicativeness and meaning as well as the development of these intuitions. First, participants in our studies were told that, despite being unfamiliar to the participant, all communicative actions were conventional. Do people hold the same expectations about novel or ad hoc communicative action? In these situations, people's expectation that communicative action should reveal non-world-directedness might be even stronger, as the lack of a convention could increase communicators' pressure to ensure that a novel gesture is recognized as such. At the same time, however, novel and ad hoc gestures must go beyond revealing their purpose (i.e., to communicate) and they must also reveal their meaning (i.e., what the gesture represents), due to the lack of a convention. This suggests that observers might expect novel communicative action to simultaneously reveal its communicative purpose and its meaning. Related research suggests that people often expect novel gestures to reveal meaning through pantomime and iconicity (e.g., if a movement looks like hammering, then it probably represents hammering; refs. [51,52]). This creates a challenge for gesturers, as their movements must simultaneously resemble world-directed action while also revealing that the movements are not actually world-directed. Gesturers might solve this problem by relying on the context to ensure that the movement cannot be incorrectly

perceived as world-directed (e.g., reaching to touch an object but stopping before actually touching it, ensuring that observers recognize the actions did not change the state of the world; refs. [53,54]). To our knowledge, however, how people balance the pressure to reveal communicative action and the pressure to make its meaning decodable remains largely an open question.

A second open question is whether the inferences we found here emerge early in development, or whether they require extended experience interpreting world-directed and communicative action. Research with infants suggests that the ability to reason about world-directed movement emerges before 12 months of age[3,6,8,26,55], and inferences about communicative action may therefore emerge at the same time. Alternatively, intuitions about communicative movements may develop as a function of experience. In particular, gestures around the world might share a common structure that helps people build these intuitions. Indeed, some conventional gestures seem to demonstrate their lack of world-directed purpose through rarity (e.g., winking, thumbs-up, okay) or repetition (e.g., waving, nodding, go away). This suggests that cultural evolution may lead to the transmission of gestures that are more identifiable as communicative, thus resulting in conventional gestures that are high in rarity.

Finally, while our work focused on explicit communicative action, our theoretical framework shares similarities with work in three adjacent fields: sensorimotor communication, ritual, and sign language. First, research in sensorimotor communication has found that people pursuing world-directed goals spontaneously introduce inefficient movements to help observers quickly identify which goal they are pursuing (e.g., curving a hand movement more than necessary to make its direction clear; refs. [48,49,56,57]). This work shows further support for the idea that not all inefficiency is communicative, as some inefficiencies serve the purpose of further establishing a world-directed goal. However, the world-directed inferences that people make from small inefficiencies, and the communicative inferences that people make from rare and repetitive actions may arise from the same inferential system, where observers analyze movements to decide if the inefficiency is designed to accentuate an intention to act on the world, or an intention to convey a message.

Second, research on ritual has argued that people expect ritualistic behavior to deviate from efficient world-directed action[58,59]. Our work is related to this proposal, but deviates in a key way. In our proposal, only certain types of action will be detected as communicative: Those that quickly and unambiguously reveal that they lack a world-directed goal. By contrast, ritual is believed to be detected by a general sensitivity to causal opacity[58–61]. That is, behavior is ritualistic when observers cannot infer an underlying cause. From this standpoint, inferences about ritual may be elicited not only when the behavior cannot be understood as world-directed, but also when the behavior cannot be understood as a communicative action, thus providing causal opacity for both instrumental and immediate social goals.

Third, beyond the type of isolated communicative action that we focused on here, communicative action can also be embedded in complete linguistic systems such as American Sign Language and Nicaraguan Sign Language[62]. How does our framework apply to communicative action in language? In the context of sign languages, the demand to continuously detect communicative action may be drastically reduced, for three reasons. First, observers in a conversation in sign language will already expect actions to be communicative. This may enable observers to treat all movements as potentially communicative and focus on recognizing each movement's meaning. Second, observers' knowledge of the sign language can further constrain the expectations about where in the body communicative action will be produced (e.g., expecting mouth morphemes in ASL; ref. [63]), alleviating the problem of revealing

which aspects of someone's body movements are communicative and which are not. Finally, observers' real-time processing of an unfolding utterance can further constrain signers' expectations about which word will follow (relying on semantic constraints; refs. [64,65]). This may allow observers to anticipate the forms of subsequent signs for quicker identification of communicative action.

The reduced expectation that every sign must reveal its communicative purpose might also provide increased flexibility in the shape that signs can take. This could provide critical degrees of freedom that are needed to produce a more expressive system where movement encodes phonological[66,67], morphological[68], and syntactic[62] structure. Similarly, the prevalence of iconicity and pantomime in sign languages (refs. [69–72]; especially in emerging sign languages[73]) suggests that signs may be subject to pressures to link form and meaning, which may also affect the shape of signs without regard to their rarity. It is possible that because of both the pressure to create linguistic structure in movement and the reduced inferential burden of detecting communicative action, sign languages may reuse features such as rarity and repetition for different purposes. For example, multiple signed languages rely on repetition (i.e., reduplication) to encode grammatical features such as switching verbs to nouns (in American Sign Language[74]), or the pluralization of certain nouns (in German Sign Language[75]; see SI for a related discussion about revealing communicative goals in vocal communication).

At the same time, this does not imply that sign languages lack the need to reveal that movements are communicative. Signs may include intrinsic flexibility that allow the introduction of ad hoc movements that help reveal communicativeness when necessary. Specifically, signs are thought to consist of three building blocks that vary in flexibility: handshape (which is highly conventionalized and least flexible), movement, and location (which is least conventionalized and most flexible, allowing signs to sustain their meaning, independent of the position in space where they are produced; refs. [76–78]). The degrees of freedom in a sign's movement and location have been hypothesized to allow signs to physically depict aspects of a scene or interaction (e.g., ref. [77]). However, it is also possible that signers use this flexibility to reveal that an action is communicative. When signers encounter ad hoc environmental constraints that may make it harder for their sign to be recognized as communicative, they could modify the sign dimensions that are least conventionalized to increase the sign's rarity. This may be especially useful in the context of iconic or pantomimic signs, which may resemble world-directed action; changing their location or movement away from objects in the environment that could confuse observers may enable signers to maintain the representational nature of the signs while also helping observers infer their communicative goal (e.g., refs. [52,53]). This view predicts that, within sign languages, observers might have a stronger expectation that position and movement are shaped to reveal non-world-directedness, relative to handshape.

Taken together, this view predicts a tradeoff between linguistic pressures and inferential demands: The expectation for communicative action to reveal its purpose should be stronger for movements that are less immersed in complex linguistic systems and movements that can be produced at any moment (rather than exclusively within complete grammatical utterances). If this is correct, then the expectation that communicative action will reveal its goal should be strongest for emblems—a type of conventional gesture that can be produced and understood in isolation, and at any point independent of the context (e.g., waving hello, or a thumbs-up; refs. [20,21]). This is a prediction that we hope to test in future work.

Altogether, our work sheds light on the interplay between gesturers' need to reveal when a movement is communicative, and the

inferential burden this imposes on their observers. Although theories of human action understanding have classically distinguished between mechanisms that support world-directed inferences and communicative inferences, both activities are generated by minds that are motivated to fulfill their goals rationally and efficiently. Our work shows that a unified expectation for rational action enables us, as observers, to recognize what other agents are doing, whether it is acting to change the physical world, or acting to communicate. As observers, holding actors to an expectation that they are efficient and rational—in both the physical and social world—structures our unique ability to build nuanced models of other people's minds from their behavior.

## Methods

**Studies 1–3, and Explanation Control (point displays).** This set includes four original studies, each one with a pre-registered replication. See SI for pre-registration details.

*Participants.* A total of 240 (30 per study) US participants (as indicated by their IP addresses) were recruited through Amazon's Mechanical Turk platform in exchange for monetary compensation based on the duration of the task. Samples of 30 participants were recruited for Study 1 (mean age = 32.87, range = 22–63), Study 1 replication (mean age = 35.40, range = 24–59), Study 2 (mean age = 34.23, range = 23–59), Study 2 replication (mean age = 36.40, range = 21–61), Study 3 (mean age = 40.83, range = 25–73), Study 3 replication (mean age = 37.80, range = 23–69), Explanation Control (mean age = 36.80, range = 21–63), and Explanation Control replication (mean age = 35.93, range = 23–50). Gender information was not recorded, but the service's gender demographics were 55% female for US participants in 2017 when data collection began[79]. Informed consent was obtained for all participants and participants were not allowed to participate in more than one study.

*Stimuli.* Each trial consisted of a seven-second video of a white dot moving on a green background (see https://osf.io/ehb48/ for videos). To make the dot's movement visually salient, we included a fading red trail that was one and a half times the length of the dot's diameter. Paths were created by combining 4 of 16 possible primitive path segments (consisting of the four cardinal directions, four diagonal directions, and eight 90-degree arc segments) in all possible combinations to create 4520 unique paths (after removing rotations and reflections of the same path shape). We sorted the paths into eight classes based on a priori features of interest that impacted how efficiently the path moved from its start to its end-point (see Fig. 1 caption for class descriptions). Four paths that captured the range of possible motions were selected as warm-up videos for all studies.

Stimuli for Study 1 and the Explanation Control were obtained by selecting the two unique maximally efficient paths from movement class A, and three random paths from each of the remaining seven movement classes, for a total of 23 paths. Stimuli for Study 3 consisted of 12 paths randomly selected from Study 1; one path was randomly selected from the eight a priori classes and four additional paths were randomly selected from the paths that received an average communicativeness rating of at least 4.25. Each of these 12 paths was presented in two ways: *bordered*, in which the path was closely surrounded on both sides by amorphous blue areas representing lakes, and *unbordered*, in which the path was not closely surrounded by lakes.

Stimuli for Study 2 consisted of 21 seven-second videos similar to those from Study 1. We selected seven basic movements from the set of paths that could be built by combining two of the sixteen possible primitives (avoiding combinations where the second primitive retraced the first; e.g., a path segment moving to the right followed by a path segment moving to the left). We then made two additional versions of each basic path: a one repetition version, in which the basic path retraced itself back to its origin, and a two repetitions version in which the basic path retraced itself back to its origin, and then repeated that path again back to its origin. To obscure the critical manipulation, paths with one repetition were rotated 90 degrees counterclockwise and reflected over the x-axis, and paths with two repetitions were rotated 180 degrees counterclockwise. The length of each path segment was adjusted so that the total distance traveled in each path was the same.

*Procedure.* Participants first read a brief cover story explaining that they would watch videos of an anthropologist moving around an island. In Studies 1–3 participants were told that the anthropologist was either completing tasks on the island, or communicating with a helicopter watching her from above. To minimize the role of prior expectations about how often an anthropologist would communicate, participants were also told that the anthropologist would be trying to communicate in roughly half of the videos. In the Explanation Control, participants were told that the anthropologist would be moving around the island, without mentioning communicative action (see SI for full cover stories and procedure details). Participants then completed a brief questionnaire to ensure they had read the instructions (see SI). Participants that answered any question

incorrectly were directed to reread the directions and attempt the survey again. Participants were not allowed to continue until they answered all the questions correctly. Participants then watched four warm-up videos in a random order. This was to ensure that participants had a sense of the range of possible paths and that they could calibrate their responses accordingly.

Participants in Studies 1–3 were asked to rate how likely they thought it was that the anthropologist was communicating with the helicopter on a Likert scale of one (*definitely not communicating*) to seven (*definitely communicating*). Participants in the Explanation Control were asked to describe one possible activity that the anthropologist was doing, and rate how difficult it was to come up with an explanation on a Likert scale of one (*extremely easy*) to seven (*extremely difficult*). Trial order was fully randomized in Studies 1, 2, and the Explanation Control. In Study 3, participants were randomly assigned to one of five trial orders that were pseudo-randomized such that the bordered and unbordered versions of the same path were never shown consecutively.

*Data analysis.* Data were analyzed using R Studio (Version 1.2.5042; ref. [80]) and R packages lmerTest (3.1–2; ref. [81]), tidyverse (1.3.0; ref. [82]), boot (3.1–25; ref. [83]), and stargazer (5.2.2; ref. [84]).

We took an effect-size estimation approach to data analyses. Rather than drawing conclusions based on significance tests, we instead estimated effect sizes with corresponding 95% bootstrapped confidence intervals. We interpret effects as reliable when a confidence interval does not cross chance performance. For completeness, we also show that our results replicate using null-significance hypothesis testing through multi-level modeling of participant judgments (see SI).

Because people expect agents to pursue goals efficiently[6,10,12,26], we quantified rarity as $r(p) = 1 - \frac{d^*(p)}{d(p)}$, where $d^*(p)$ represents the shortest distance between the beginning and end of a path and $d(p)$ represents the actual distance traveled. Thus, $r(p) = 0$ when the observed path is maximally efficient (and therefore expected for world-directed goals). $r(p)$ increases as the path includes additional movements that are not directed toward efficiently reaching a final destination.

**Studies 4–7 (naturalistic videos).** Studies 4, the Familiarity Control, and 5 were pre-registered, but the Weirdness Control was not. Due to time-pressures associated with cross-cultural research, the Tsimane' samples for Studies 6–7 (which were collected prior to US samples) were not pre-registered. Sample sizes were determined by the number of conditions in a study (to ensure that each trial had sufficient data) and time constraints associated with cross-cultural research. See SI for details.

All studies with Tsimane' participants were conducted with a local interpreter. All task paradigms and instructions were discussed in advance with the interpreters to ensure clarity and fluidity. Specifically, all Tsimane' studies were designed with feedback from Tsimane' interpreters to ensure that the cover stories, critical task elements (e.g., the idea of communicating through movements that might look different than conventional gestures used by the participants), and test questions, were clear and that their phrasing did not have any unforeseen implications arising from translation. Participants were allowed to participate in no more than one study.

*Local approval, consent, compensation, and debrief procedures with the Tsimane'.* All studies conducted with the Tsimane' were approved locally by the *Gran Consejo Tsimane'* (Grand Tsimane' Council, the main political body in the Tsimane'; GTC), and from the individual communities where we conducted our studies, in line with local norms and preferences. Prior to all research, the first and last two authors met with the council to discuss our research questions, experimental procedures, type of data to be collected, potential benefits to the Tsimane' communities, potential negative consequences, and compensation to participants. Having secured unanimous approval from the GTC, we also obtained approval from each individual community that we visited, and from each participant (see SI for full consent process).

The compensation scheme for participants and communities was determined in conjunction with the GTC, which included material gifts for each participant with an approximate value of $7 USD, a donation to the Tsimane' communities (administered by the council) to support local education and health initiatives, and a soccer ball for each visited community.

Details about the purpose of the study and its outcomes were both explained to the GTC and to the communities. The researchers and interpreters were also available to answer any questions both before and after data collection. In addition, interpreters conferred with the local teachers (in communities where they were present during testing), who were made aware of the purpose of the study to facilitate internal transmission of study purposes. At the conclusion of our trip, AR, TH, and JJE met with the GTC and additional members of the community (invited by the GTC) to share the outcomes of our studies.

*Participants.* Study 4, Familiarity Control, Weirdness Control: 150 US participants were recruited through AMT. Thirty participants completed each condition (Study 4 low punctuality, mean age = 38.40, range = 23–60; Study 4 natural punctuality, mean age = 34.17, range = 23–58; Weirdness Control, mean age = 37.73, range = 23–60; low punctuality Familiarity Control, mean age = 36.8,

range = 22–68; and natural punctuality Familiarity Control, mean age = 35.70, range = 20–60).

Study 5: 40 US participants were recruited from the Yale University undergraduate subject pool, 40 San Borjan participants (mean age = 31.78, range = 16–63, 26 female participants) were recruited in San Borja, Bolivia, and 180 Tsimane' participants (mean age = 31.97, range = 14–71, 122 female participants) were recruited in their communities. 120 of these Tsimane' participants completed the low-punctuality condition and 60 completed the natural-punctuality condition (reduced based on a power analysis using the data from the low-punctuality condition). 58 additional Tsimane' participants were recruited, but excluded from the study (see SI for details and exclusion criteria).

Study 6: 59 US participants (mean age = 36.39, range = 23–71) were recruited through AMT and 59 Tsimane' participants (mean age = 30.86, range = 15–64, 35 female participants) were recruited in their local communities. 31 additional participants were recruited but excluded because they gave the same response for all stimuli (5 US and 26 Tsimane').

Study 7: 100 US participants (mean age = 41.52, range = 20–77) were recruited through Amazon's Mechanical Turk platform and 32 Tsimane' participants (mean age = 34.88, range = 19–73, 24 female participants) were recruited from their local communities. One additional Tsimane' participant was recruited but excluded from the study due to an experimenter error.

*Stimuli.* Stimuli consisted of 24 videos showing different body movements. This set consisted of 6 basic motions that could be confounded with a world-directed goal, and three modifications per video: a rare version, a repetitive version, and a rare + repetitive version (Fig. 3). Each movement was taped with an actor who was blind to the purpose of the study and its relation to communication and gesture (natural punctuality videos), and with a knowledgeable actor who explicitly avoided making punctuated changes in speed and velocity that are typically associated with gesture (low punctuality videos). Both actors timed their movements with a metronome, ensuring that the movements were done at a consistent speed.

The demonstrator was only visible from the mouth down to the table in front of her, at the level of her abdomen. Before and after making each movement, the woman stirred the contents of a bowl sitting on a table in front of her in order to clearly bookend the relevant motions that participants would judge. The movements were then looped to create 2-min videos. All movement types were normed to ensure they captured the critical properties. See SI.

*Procedure.* Study 4: Participants read a cover story explaining that they would see videos of a person who is sometimes using her hands to communicate with her friend. The cover story explained that the person was from a remote island (so that participants would not expect to see familiar communicative gestures) and that she would be trying to communicate in roughly half of the videos (see SI for full cover story and procedure details). Participants then completed a three-question survey to ensure they understood the task. If participants answered any question incorrectly, they were directed to reread the directions and attempt the survey again. Participants were not allowed to continue until they answered all the questions correctly. Participants were randomly assigned to one of three trial orders in which the 24 videos from either the low or natural punctuality video set were pseudo-randomized such that none of the variations of the same basic movement were presented consecutively. In each trial, participants rated how likely they thought it was that the person was communicating on a Likert scale of one (*definitely not communicating*) to seven (*definitely communicating*).

Weirdness Control: Participants read a cover story similar to the one from Study 4. For each video, participants were asked to rate how weird her movements were on a scale of one (*Not weird*) to seven (*Extremely weird*). See SI for full details.

Familiarity Control: Participants read a cover story similar to the one from Study 4 with the difference that they were told that the person was always communicating. For each video, participants indicated whether they recognized the gesture on a scale of one (*I definitely do not recognize it*) to seven (*I definitely recognize it*), write what they thought the gesture meant, and rate how confident they were that their previous response was the actual meaning of the gesture on a scale of one (*not confident at all*) to seven (*extremely confident*). See SI for full details.

Study 5: Study 5 was an adaptation of Study 4 for testing with the Tsimane'. Each trial consisted of a pair of videos, one basic video and either its corresponding rare version or its repetitive version, stacked vertically on an iPad. Participants were told that they would be shown two videos of a person moving and that their task was to indicate in which of the two videos the person was communicating.

To ensure participants understood the task, they first completed two warm-up trials which contrasted novel movements against familiar communicative gestures that we confirmed were well known among all three groups (hi and no). In each warm-up trial, one video showed a rare and repetitive novel movement, and the other video showed the demonstrator waving their hand, or shaking their head (warm-up trial order, and video presentation order within each trial counterbalanced across participants). Participants were then asked to identify the video where the person was saying hi or no (depending on trial). If participants answered incorrectly, the experimenter prompted participants to produce the gesture associated with the message, showed the videos again, and then asked the question once more. Participants who answered either one of the warm-up questions incorrectly twice were excluded from further participation in the study.

The warm-up questions helped us to highlight the goal of the task, while also biasing participants against our predicted results (since participants had to refrain from choosing the novel rare and repetitive movements in order to pass the warm-ups).

After the warm-up, participants were told that they would now watch videos of a person from very far away who does not use gestures familiar to the participant (See SI for full script). The actor from the test videos was different from the one in the warm-up videos. Participants were then shown six pairs of videos. Three pairs contrasted a basic movement against the corresponding rare movement, and the other three pairs contrasted a basic movement against the corresponding repetitive movement. Participants never completed more than one trial with each basic movement type, and stimuli was presented in one of four pseudo-randomized orders such that no more than two of the same video contrast type were presented consecutively (e.g., basic vs. repetitive or basic vs. rare).

Study 6: US participants completed the study on their home computer. Tsimane' participants were read the instructions by a translator and videos were presented on an iPad (see SI for translated script and full procedure). Participants learned that they would watch videos of someone producing a body movement and that their task was to determine if the person was alone in the room or whether someone else was in the room with her. Participants watched two warm-up videos (presentation order counterbalanced). In one video, a demonstrator stopped mixing the bowl to point at something, and participants were told that the pointing suggested that there was someone else in the room. In the other video, a demonstrator stopped mixing and clapped sharply in mid-air. Participants were told that the demonstrator had killed a fly, and they might be alone in the room. Finally, participants were told that there would be another person in the room in about half of the videos.

The test trials consisted of the rare + repetitive videos from the low punctuality video set from Study 4 (using a different actor than the one shown in the warm-up videos). Videos were presented one at a time and Tsimane' participants could request to watch the video again. Each participant was randomly assigned to one of four trial orders. Each trial order had three basic movement videos and three rare + repetitive videos pseudo-randomized such that two videos of the same movement type were not presented consecutively and that the participant did not see both versions of the same movement.

Study 7: US Participants completed the study on their home computer. Tsimane' participants were read the instructions by a translator and videos were presented on an iPad (see SI for translated script and full procedure). Participants were introduced to the videos in a similar way to Study 6, except that they were now told that they would have to infer what the person was doing when they stopped cooking (without mentioning the presence or absence of any other agents). To encourage participants to explain the movements in terms of goals rather than describe the low-level features of the movements, participants were shown the same two warm-up examples from Study 6 (presentation order counterbalanced) and received an explanation for each video (explaining that in one video the person pointed to something and in the other, the person killed a fly). Participants then watched the same low punctuality rare + repetitive videos from Study 6. Each participant was randomly assigned to one of four trial orders from Study 6.

*Data analysis.* Data were analyzed using the same statistical tools described for Studies 1–3. Our approach to data analyses was the same as in Studies 1–3, with the difference that we also relied on permutation tests to estimate $p$-values associated with participant distribution of responses. See SI for analyses pre-registrations and deviations from pre-registered plan.

*Compliance with ethical regulations.* All studies detailed here received ethical approval from the Yale Human Subjects Committee and complied with all relevant ethical regulations. All studies conducted with the Tsimane' also received local approval from the Grand Tsimane' Council. Informed consent was obtained from all participants.

**Reporting summary.** Further information on research design is available in the Nature Research Reporting Summary linked to this article.

## Data availability
All associated pre-registrations, stimuli, and datasets are publicly available on the Open Science Framework. Files from Studies 1–3 and the Explanation Control are available at https://osf.io/ehb48/. Files from Studies 4–7 are available at https://osf.io/wxdka/. Figures 2, 5, 6 and 7 were created using data from their corresponding studies as indicated in the figure captions.

## Code availability
All data analysis code from Studies 1–3 and Explanation Control is available at https://osf.io/ehb48/. Data analysis code from Studies 4–7 is available at https://osf.io/wxdka/.

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

## Acknowledgements

We thank Madison Flowers for help with piloting Studies 1–3. We thank Esther Conde for logistical support in conducting the cross-cultural work, and Manuel Roca and Elias Hiza for help translating and running the tasks with Tsimane' participants. We thank Gergely Csibra, Sammy Floyd, Padraic Monaghan, Paula Rubio-Fernandez, and Marieke Schouwstra for useful comments on this work. This work was supported by supported by the US National Science Foundation grant BCS-2045778 awarded to J.J.-E.

## Author contributions

A.R, J.J.-E, and R.A. conceptualized Studies 1–3. A.R. and J.J.-E. conceptualized Studies 4–7, in consultation with T.H. A.R. designed the stimuli, collected, and analyzed the data for all studies under the supervision of J.J.-E. A.R., J.J.-E., and A.C. conceptualized the computational model, and A.C. implemented it under the supervision of J.J.-E. T.H. coordinated researcher with the Tsimane' communities. A.R. wrote the manuscript with J.J.-E. A.C., R.A., and T.H., provided critical comments. All authors approved the final version of the manuscript.

## Competing interests

The authors declare no competing interests.
