## [Peer Review File · Nature Communications]

People infer communicative action through an expectation for efficient communicationREVIEWER COMMENTS

Reviewer #1 (Remarks to the Author):

The current manuscript puts forward a novel proposal regarding how people identify communicative actions. Building on Bayesian models of goal inference, and the foundational assumption that people act efficiently to achieve their goals, the manuscript proposes that actions will be seen as communicative when they efficiently show that they lack an external goal — that is, when the action is not better explained through goals like manipulating an object or reaching a certain location. The paper combines a computational model with extensive data from both US participants and participants in the Tsimane tribe (as well as control participants from a nearby town, who are less isolated/ more familiar with Western culture).

In general, I have a favorable view of the theory put forward in the paper. The logic of the experiments is clear, and I was impressed by the broad range of inferences this process is shown to support, in line with their broader account (e.g. in addition to the predictions of the parametrically-varying trials based on the model, also inferring whether there is a second person present in the room, and providing written explanations of the actors' goals that were in line with predictions).

The particular experiments have the potential to knit together a broad set of literatures in goal inference and action concepts, ritual, and gesture. However, the current version of the manuscript does not maximize this potential, in several ways (detailed below) which could potentially be addressed in a revision. I recognize that current limitations in deeply integrating literature are, in part, due to length limitations. It is possible that this issue could be addressed in a careful revision in this format and in this outlet. However I think an expanded version of this manuscript in a longer-form outlet would also be highly influential, and perhaps clearer for the reader.

1) For example, a core issue in the paper is how people reason about repetitive movements. This relates deeply to literatures on ritual (e.g. work by Cristine Legare and colleagues), which introduces a related distinction between interpreting behavior as instrumental (equivalent to world-directed) versus conventional (another social purpose, though not the same as communication). This distinction (and the detection of ritual) is also thought to be driven by noticing that actions did not enact any change in the world. It would be relevant and informative for the paper to discuss how the current proposal differs from and relates to this work from the literature on ritual. When will we see actions as communicative versus conventional — or are these overlapping sets, given that we often use conventional actions to communicate our social identities? How should we put these ideas together? There are also interesting other literatures with repetitive communicative signals, like reduplication in linguistics (very, very, very small; used for emphasis).

2) Along similar lines - the naturalistic stimuli that resemble sign language would benefit from more connection to the literature on gesture and sign language. With regard to gesture — work by Miriam Novack is likely of interest, as it uses a similar goal inference framework to think about when we categorize movements as gestures (e.g. Novack Wakefield & Goldin Meadow 2016; Wakefield Novack Goldin-Meadow 2018). With regard to sign language - How do babies recognize sign language as communicative? This question seems like it could provide useful context re: how adults would recognize these gestures as communicative in nature (e.g. Ferguson & Lew Williams 2016, & maybe work by Rain Bosworth on attention to ASL in infancy, may be relevant here).

3) I was somewhat surprised by the finding (and the theoretical argument) that participants did not find it difficult to construe the rare, repetitive actions as world-directed. The logic of this theoretical prediction should be unpacked. E.g. perhaps the thinking is: if people could not construe these actions as world-directed, perhaps they could be using a simpler heuristic rather than a rational inference? However it seems to me that this finding is not an obvious prediction of the Bayesian model. In particular, the Bayesian inference people are making can be stated as: Which goal is the best

explanation for this behavior? A physical-world-related explanation? Or communication? So, in order to come to the conclusion that communication is the best explanation, one must think that the actions are less well-explained by (or less consistent with) physical-world-related goals. In the extreme, people thus should find it harder to explain these actions through physical-world-related goals — or, if not harder, at least they should find these explanations less compelling or convincing. In any case - it seems necessary to unpack the prediction regarding how hard or easy it should be to construe rare/repetitive actions as having world-directed goals, and how it falls out of the model.

4) Some of the work cited showed that when movements are inefficient as a means to any object or location, including (but not limited to) repetitive movements, people often infer that the goal was to produce the particular movements, for their own sake (rather than for any physical-world-directed goal - Schachner & Carey 2013). How should the reader think about the relationship of those findings to these? It seems like the current model would predict that the stimuli in this older work should be seen as communicative.

This points to one of the most complex issues of action representations: They are hierarchical. That is, a single action is seen as having multiple goals, at different levels of abstraction. Thus, a single reaching action may have the goal of "grasping a handle" and also the goal of "drinking coffee" and also the goal of "becoming more awake" and also the goal of "finishing this manuscript". One possibility is that the actions under investigation here, as well as those in the work by Schachner & Carey, are an overlapping set. In other words, an inefficient, repetitive action may be seen as having the goal of performing those particular repetitive movements (and thus what they term a movement-based goal) - but at the same time, at a more abstract level, have the goal of getting an observer to recognize that they are trying to communicate. This is one possible interpretation — there are other very reasonable interpretations as well. However, it would be useful to provide an account that more deeply integrates these findings. (This integration, along with the below suggestions for links to other accounts, would serve to flesh out the claim that "not all deviations from world-directed actions are communicative" (page 22 1st paragraph), a line with important implications which should be more fully developed.)

5) I suggest that the terminology used should be modified, to help the reader hone in on the crux of the issues. Particularly, with regard to the terms "world directed" and "rare".

5a) The term "world-directed". This refers particularly to interactions with the physical world (as in the example provided: reaching for and manipulating objects). This is contrasted with communication. While this conceptual distinction is fine and useful to make, the term world-directed does not do justice to this distinction. After all, other people are part of the world as well — the social world. Communication is an act meant to have an impact on other people (and thus the world). Thus the distinction in the paper may be better labeled as directed toward the physical world versus the social world (changing to physical-world directed vs. communication would be enough to make this clearer, perhaps).

5b) The paper uses the term "rare", with the claim that rare actions are seen as more likely communicative. Rare implies unusual, or not typically produced. However what is termed rarity is actually quantified in the paper as inefficiency — "how much each movement diverged from the shortest path". Why say rarity when what is meant is inefficiency toward physical-world-directed goals?

In my view using the term "inefficiency" rather than rarity would help to clarify the relationship of this work to prior work in action understanding, in a useful way. For observers, inefficiency toward other goals provides powerful evidence for exactly what the Bayesian model in the paper instantiates — that the actions are not well-explained by the whole class of physical-world-directed goals.

Minor comments:

Page 18 paragraph 1 last line: is this percentage the percent reporting another person is present, or absent? This is currently ambiguous based on the surrounding text.

P.11 Paragraph 3 - here the prediction of the account is stated, but in this context this prediction needs to be unpacked. I suggest explaining or briefly stating how this prediction falls out of the model, at this point in the paper.

p. 7 line 8, need a semi-colon before Fig.2a

p. 8 line 7, Fig.1d should be Fig. 2d

Reviewer #2 (Remarks to the Author):

This paper reports a series of elegant experiments that advance our understanding of communication. Overall, I am enthusiastic about the paper. Both of the novel tasks designed to study expectations about communication are innovative and well-controlled, providing a substantial empirical contribution. The paper is also making a number of useful conceptual contributions to the literature and will likely have impact on future studies of communication and action understanding. However, there are several issues, including questions concerning the generalizability of the results, that I would like to see addressed in a revision.

Major points:

1. "Rarity" should be defined early on in the paper and it should be made clear how the concept of rarity differs from efficiency.

2. As far as I understand, participants in all studies were made to believe that if there was any communication, it was based on a previously established, conventionalized, communication system. So, when participants performed the task of judging whether observed actions were or were not communicative, they were judging whether observed actions were part of a conventionalized communication system, rather than judging whether someone was trying to communicate impromptu in the absence of an established system. If my understanding is correct, then I would find it important that the authors make this point more explicit in the paper and discuss whether they would expect the results to generalize to a version of the experiments that does not mention a previously established code in the instructions. I find this important because people might have different assumptions about established communicative actions versus non-conventionalized communicative actions.

3. The authors seem to assume that instrumental goals (or world-directed goals in their terminology) are performed most efficiently without repetition and by minimizing distances to be covered. However, there are instrumental goals, like searching for something or exploring a space, that involve repetitions and necessary deviations from the shortest path. For example, if participants in Study 1-3 had been told that the anthropologist is either communicating or searching for her glasses that she has lost, the results might be quite different. It seems important to acknowledge that the distinction between instrumental and communicative actions is not always as clear-cut as the present studies suggest, and to discuss whether the results would generalize to situations where instrumental actions include exploration or searching behavior.

4. The explanation control results are not fully convincing, as the judgments are consistently around 4. (How) can the possibility be excluded that the question just did not make much sense to participants? To get at this, can the authors report the variability for each data point in Fig 2 d)?

5. Concerning the model predictions for Study 1-3, how does the model fit change when the two movement classes with the largest discrepancy between model and participants (B and D in Fig 3 c))

are taken out? Does this significantly improve the model fit?

6. In Studies 4-7, the "world-directed" actions could also be described as "body-directed". There is research in cognitive neuroscience suggesting that actions performed on the body/tools used on the body/words for actions and tools used on the body are represented in a different way than actions/tools/words for actions and tools use to act on the outside world, so it might be worth discussing the status of body-directed actions compared to actions performed on the environment.

Minor:

There is a numbering problem in the list of references: Reference 20/21 list 4 rather than 2 references.

p. 18 last sentence is not clear ("Among the subset...")

Reviewer #3 (Remarks to the Author):

This paper presents a series of studies of movements showing that certain characteristics of movements (rarity operationalised as deviation from direct paths between points, and repetition) relate to movements being more likely to be classified as communicative. The work presents a thoroughly conducted set of studies, testing these hypotheses about movements in different ways (movements on a map, and hand gestures) and on different populations. It is an impressive quantity of work. The studies have the potential to be of broad interest in terms of highlighting aspects of gesture and action that suggest communicativity of the signal to another, and my sense is that they would generate substantial interest over studies of communicative systems.

There are a few issues that need to be strengthened to maximise the impact of the work presented in this paper.

First, iconicity is a key issue for sign languages (and spoken languages) but it is relatively sidelined in the paper (there are some references to work on this on p.22). As these references (and others - see work for instance by Vinson, Vigliocco, Perniss) point to iconicity as being hugely important as a characteristic of sign language systems, this needs to be foregrounded more. In particular, the definition of communicative is often characterised as non-goal directed, and this limits the way in which communicative can be interpreted (by the participants in the studies, and by the readers of the article). I would like to see greater discussion of iconicity and how rarity and repetition might relate to iconicity in sign systems. Iconicity seems to fall somewhat between goal-directed and non-goal-directed. For instance, in BSL the sign for eat is close to a mime to eat (but repeated) <https://www.britishsignlanguage.com/bsl-dictionary/eat/> . Is that goal-directed, or communicative, or both goal-directed and communicative? That is not yet clear. If the paper is dealing only with aspects of communication that are not iconic then that needs to be stated. If the paper is addressing iconic signs as well as non-iconic signs then that needs to be outlined more clearly how that is the case. The discussion on p.23 seems to suggest iconicity is another aspect of signs in addition to rarity/repetition cues, but I think it might be very difficult to isolate it from the fundamental nature of signs in human communication.

Second, the authors include an additional classification in Study 7 as "descriptive". In the main paper analyses, descriptive responses were omitted, but they comprise a large proportion of responses, particularly in the US sample (Supp mats p.21, show 32% of basic movements were classified as descriptive and 33.3% of rare and repetitive movements). These are stable proportions across the types of movements and so the comparison between world-directed and communicative across movement types is not necessarily unduly affected, but how to take them into account? It could be that "descriptive" is closer to what other studies intend by "iconicity" in signs. In this case, there may

be some difficulties in characterising signs as descriptive but not communicative. To whom is the signer describing if not another person? This descriptive signification seems like a key part of definition of what it means to be a sign and so it is odd to distinguish it from other communicative goals. Again, the authors need a more careful treatment and justification for the distinction between communicative and descriptive, how participants will have interpreted that distinction, and how that relates to human communicative systems.

Third, the study omits mention of speech as a system of communication. But as this is the primary form of communication between humans it needs to be linked to the content of the paper more closely, otherwise the paper risks being seen as addressing only non-oral communication, which then limits the broader implications of the work. Should we expect speech to be (maximally) distinct from non-communicative oral productions, should we expect it to contain repetitions, or will it not fall under the same principles? If not, why not? As an example of how speech and non-speech communication can be compared, Perlman et al. (2018) showed that speech and sign languages can both be considered in terms of iconicity and there are parallels in terms of where iconicity is found, for different grammatical categories for instance, but in the current study it is hard to translate (excuse the term) ideas contrasting goal-directedness with rarity in the speech domain (there might be some mileage in repetition - a hallmark of child-directed speech for instance (though within words (syllables or phonemes) it is rarer than one might expect by chance).

Small points:

pp.23-24 "Our capacity to recognize even novel movements as communicative suggests that the expectation for efficient action has broader scope than previously believed. And yet, this expectation may be more powerful than what our work captures, perhaps supporting the recognition of an even broader suite of human behaviors such as epistemic actions (like reading a book or exploring one's surroundings) and challenging actions which are their own reward (like mastering a skill or solving a puzzle)." I didn't understand how the expectation for efficient action might provide insight into actions like reading a book or exploring surroundings. This could do with more exposition to make the link clearer, or omitting if it is only a tenuous link.

p.28 In the data analysis for quantification of rarity, can the authors indicate to what extent rarity is independent of distance? A small deviation from shortest distance for a short route would result in a higher rarity value than a small deviation from shortest route for a longer route, based on how rarity is calculated. Could overall route length be a factor in affecting decisions in the studies?

Reference

Perlman M, Little H, Thompson B, Thompson RL. Iconicity in Signed and Spoken Vocabulary: A Comparison Between American Sign Language, British Sign Language, English, and Spanish. *Front Psychol.* 2018 Aug 14;9:1433. doi: 10.3389/fpsyg.2018.01433.

Reviewed by Padraic Monaghan (signed review)

Response to Reviewer Comments.

We are grateful to the reviewers for providing us detailed feedback on our manuscript.

We have worked to address each and every concern and present a point-by-point response below. To make this letter easier to read, we present all reviewer comments in blue 12pt font, all of our responses in black 12pt font, and all manuscript quotes in black 10pt font with a one-inch indentation. In some cases, we found it most natural to split a reviewer comment into multiple responses to discuss how we addressed each sub-component of a concern. Because this led to inconsistencies in the numbering system, we have re-labeled all reviewer points using label X.Y for Reviewer X's point Y. We use this system throughout to direct reviewers to related comments.

Reviewer #1 (Remarks to the Author):

1.1. The current manuscript puts forward a novel proposal regarding how people identify communicative actions. Building on Bayesian models of goal inference, and the foundational assumption that people act efficiently to achieve their goals, the manuscript proposes that actions will be seen as communicative when they efficiently show that they lack an external goal — that is, when the action is not better explained through goals like manipulating an object or reaching a certain location. The paper combines a computational model with extensive data from both US participants and participants in the Tsimane tribe (as well as control participants from a nearby town, who are less isolated/ more familiar with Western culture).

In general, I have a favorable view of the theory put forward in the paper. The logic of the experiments is clear, and I was impressed by the broad range of inferences this process is shown to support, in line with their broader account (e.g. in addition to the predictions of the parametrically-varying trials based on the model, also inferring whether there is a second person present in the room, and providing written explanations of the actors' goals that were in line with predictions).

The particular experiments have the potential to knit together a broad set of literatures in goal inference and action concepts, ritual, and gesture. However, the current version of the manuscript does not maximize this potential, in several ways (detailed below) which could potentially be addressed in a revision. I recognize that current limitations in deeply integrating literature are, in part, due to length limitations. It is possible that this issue could be addressed in a careful revision in this format and in this outlet. However I think an expanded version of this manuscript in a longer-form outlet would also be highly influential, and perhaps clearer for the reader.

Thank you for the time invested in evaluating our manuscript and providing us with comments. We are grateful that the reviewer sees our paper as a novel contribution to the literature with the potential to be influential, and we appreciate their encouragement to connect our framework to related research areas. In the points below, we discuss the changes we have taken to address each concern.

1.2. For example, a core issue in the paper is how people reason about repetitive movements. This relates deeply to literatures on ritual (e.g. work by Cristine Legare and colleagues), which introduces a related distinction between interpreting behavior as instrumental (equivalent to world-directed) versus conventional (another social purpose, though not the same as communication). This distinction (and the detection of ritual) is also thought to be driven by noticing that actions did not enact any change in the world. It would be relevant and informative for the paper to discuss how the current proposal differs from and relates to this work from the literature on ritual. When will we see actions as communicative versus conventional — or are these overlapping sets, given that we often use conventional actions to communicate our social identities? How should we put these ideas together?

Thank you for bringing this literature to our attention. The reviewer is correct that the literature on ritual also posits that non-world-directedness (or *causal opacity* in the terminology of Legare and colleagues) is a central feature that reveals ritualistic behavior (Kapitány & Nielsen, 2015; Legare, Wen, Herrmann, & Whitehouse, 2015).

Our proposal for recognizing communicative action differs from the proposal for recognizing ritual in two important ways. First, as we understand it, the research on ritual posits that any causal opacity will trigger a ritualistic inference, whereas we propose that only certain types of inefficiencies (namely those that quickly and unambiguously reveal their lack of physical world-directness) will trigger a communicative inference (see also our response to point 5b below, and to Reviewer 2's point 1).

Second, the literature on ritual also emphasizes the importance of additional social cues. For instance, Clegg & Legare (p.527, 2016) write: "What distinguishes instrumental from ritual practices often cannot be determined directly from the action alone, but requires interpretation by the learner based on relevant social cues and contextual information"

Our revised manuscript now includes a paragraph discussing the relationship between our theoretical framework, and theories in related areas, including ritual. Here, we now highlight the similarities and differences between our account and accounts developed to understand ritual (key changes related to ritual underlined only in this quote):

While our work focused on explicit communicative action, our theoretical framework shares similarities with work in several adjacent fields. First, research in sensorimotor communication has found that people pursuing world-directed goals spontaneously introduce inefficient movements to help observers quickly identify which goal they are pursuing (e.g., curving a hand movement more than necessary to make its direction clear; 44-45, 48-49). This work shows further support for the idea that not all inefficiency is communicative, as some inefficiencies serve the purpose of further establishing a world-directed goal. However, the world-directed inferences that people make from small inefficiencies, and the communicative inferences that people make from rare and repetitive actions may arise from the same inferential system, where observers analyze movements to decide if the inefficiency is designed to accentuate an intention to act on

the world, or an intention to convey a message. Second, our theoretical framework is also related to research on ritual, which has argued that people expect ritualistic behavior to deviate from efficient world-directed action (50-51). Our work is related to this proposal, but deviates in a key way. In our proposal, only certain types of action will be detected as communicative: Those that quickly and unambiguously reveal that they lack a world-directed goal. By contrast, ritual is believed to be detected by a general sensitivity to causal opacity (50-53). That is, behavior is ritualistic when observers cannot infer an underlying cause. From this standpoint, inferences about ritual may be elicited not only when the behavior cannot be understood as world-directed, but also when the behavior cannot be understood as a communicative action, thus providing causal opacity for both instrumental and immediate social goals.

We also thank the reviewer for raising the question of how we conceptualize the relationship between communicative and conventional action. The reviewer is correct that we see these as overlapping sets: a large set of communicative action is conventional (e.g., waving hi). Indeed, participants in our studies were told that the potential communicative moments they would see were conventional (although novel to the participant). Reviewer 2 raised a very similar concern and we present a detailed response in responses R2.3.

1.3. There are also interesting other literatures with repetitive communicative signals, like reduplication in linguistics (very, very, very small; used for emphasis).

Thank you for this comment. We agree that reduplication is an interesting aspect of communication—both spoken and signed. We have reviewed the literature on reduplication and, as we understand it, this phenomena is a feature of full-blown linguistic systems (as opposed to isolated communicative signals). This helped us realize that our manuscript was vague about the scope of our theoretical proposal and how its implications are different for isolated signs relative to signs within full signed and spoken languages (a point that Reviewer 3 also raised).

We have made multiple major changes to our manuscript so that it has a clearer delineation of an analysis of communicative action in the case of isolated signs, and in the case of full-blown languages. The full list of edits can be found in responses R3.2 and R3.4.

Our revised manuscript now explicitly discusses reduplication in our general discussion, where we discuss our framework in the context of sign languages. We argue that in the context of complete linguistic systems, the pressure for every single sign to reveal it is communicative is reduced, and repetition therefore fulfills a variety of other expressive purposes, likely as a result of the pressures to form a productive, expressive, and learnable language—pressures that are not shared by isolated communicative gestures like that we expose participants to in our study.

In the Discussion (pages 26-27), after explaining how the formation of longer, multi-sign utterances in sign languages may reduce the burden for each individual sign to reveal that it is communicative, we write:

The reduced expectation that every sign must reveal its communicative purpose might also provide increased flexibility in the shape that signs can take. This could provide critical degrees of freedom that are needed to produce a more expressive system where movement encodes phonological (63-64), morphological (65), and syntactic (59) structure. Similarly, the prevalence of iconicity and pantomime in sign languages (66-69; especially in emerging sign languages; 70) suggests that signs may be subject to pressures to link form and meaning, which may also affect the shape of signs without regard to their rarity. It is possible that because of both the pressure to create linguistic structure in movement and the reduced inferential burden of detecting communicative action, sign languages may reuse features such as rarity and repetition for different purposes. For example, multiple signed languages rely on repetition (i.e., reduplication) to encode grammatical features such as switching verbs to nouns (in American Sign Language; 71), or the pluralization of certain nouns (in German Sign Language; 72; see SI for a related discussion about revealing communicative goals in vocal communication).

We also discuss reduplication in spoken languages in a SI section titled “Vocal Communication and the Pressure to Reveal Communicative Goals” (relevant text underlined):

Similarly, we also do not predict that repetition or other features would be built into complete spoken languages in order to differentiate speech from non-communicative vocalizations. While repetition is a notable form of non-arbitrariness in both signed and spoken languages, its purpose is not to signal that speech is communicative, but rather it is often used iconically to connect lexical items to their meanings [18]. However, repeated elements (i.e., reduplication) are common in baby talk (e.g., night-night and choo-choo) and these reduplicated words are easier for infants to learn [19,20]. It has been proposed that these reduplicated baby talk words “...may be more likely to be noticed in the input and stored in verbal memory than their adult-like counterparts (e.g., *train* and *good night*), making them accessible targets for initial word learning” [21, p.1979]. Why reduplicated elements have an early learning advantage is still an open question that could relate to the identifiability of reduplicated baby talk words as discrete communicative signals.

Additional details about this SI section can be found in our response R3.4.

1.4. Along similar lines - the naturalistic stimuli that resemble sign language would benefit from more connection to the literature on gesture and sign language. With regard to gesture — work by Miriam Novack is likely of interest, as it uses a similar goal inference framework to think about when we categorize movements as gestures (e.g. Novack Wakefield & Goldin Meadow 2016; Wakefield Novack Goldin-Meadow 2018). With regard to sign language - How do babies recognize sign language as communicative? This question seems like it could provide useful context re: how adults would recognize these gestures as communicative in nature (e.g. Ferguson & Lew Williams 2016, & maybe work by Rain Bosworth on attention to ASL in infancy, may be relevant here).

Thank you for this comment. We have thoroughly revised and extended the discussion section to address the connections between our work and research in ASL, discussing how our framework applies to these questions. First, when discussing open questions, we write:

Our work also opens two further questions regarding humans' inferences about communicative goals. First, participants in our studies were told that all communicative actions being produced were conventional (although unfamiliar to the participant). Do people hold the same expectations about novel or ad-hoc communicative action? In these situations, people's expectation that communicative action should reveal non-world-directedness might be even stronger, as the lack of a convention could increase communicators' pressure to ensure that a novel gesture is recognized as such. At the same time, however, novel and ad-hoc gestures must go beyond revealing their purpose (i.e., to communicate) and they must also reveal their meaning (i.e., what the gesture represents), due to the lack of a convention. This suggests that observers might expect novel communicative action to simultaneously reveal its communicative purpose and its meaning. Related research suggests that people's inferences about the meaning of novel gestures are commonly grounded in an expectation for pantomime and iconicity (e.g., if a movement looks like hammering, then it probably represents hammering; 54-55). This appears to create a challenge for gesturers, as their movements must often resemble world-directed action while also revealing that they are not world-directed action. Recent research into gesture provides one potential solution: Gesturers might solve this problem by relying on the context to ensure that the movement cannot be incorrectly perceived as world-directed (e.g., reaching to touch an object but stopping before actually touching it, ensuring that observers recognize the actions did not change the state of the world; 56-57). To our knowledge, however, how people balance the pressure to reveal communicative action and the pressure to make its meaning decodable remains largely an open question.

We then connect our work more directly to this research, and discuss the limitations of our framework with respect to sign languages:

Here we focused on isolated communicative action, but communicative action can also be embedded in complete linguistic systems such as American Sign Language and Nicaraguan Sign Language (59). How does our framework apply to communicative action in language? In the context of sign languages, the demand to continuously detect communicative action may be drastically reduced, for three reasons. First, observers in a conversation in sign language will already expect actions to be communicative. This may enable observers to treat all movements as potentially communicative and focus on recognizing each movement's meaning. Second, observers' knowledge of the sign language can further constrain the expectations about where in the body communicative action will be produced (e.g., expecting mouth morphemes in ASL; 60), alleviating the problem of revealing which aspects of someone's body movements are communicative and which are not. Finally, observers' real-time processing of an unfolding utterance can further constrain signers' expectations about which word will follow (relying on semantic constraints; 61-62). This may allow observers to anticipate the forms of subsequent signs for quicker identification of communicative action.

The reduced expectation that every sign must reveal its communicative purpose might also provide increased flexibility in the shape that signs can take. This could provide critical degrees of freedom that are needed to produce a more expressive system where movement encodes phonological (63-64), morphological (65), and syntactic (59) structure. Similarly, the prevalence of iconicity and pantomime in sign languages (66-69; especially in emerging sign languages; 70) suggests that signs may be subject to pressures to link form and meaning, which may also affect the shape of signs without regard to their rarity. It is possible that because of both the pressure to create linguistic structure in movement and the reduced inferential burden of detecting communicative action, sign languages may reuse features such as rarity and repetition for different purposes. For example, multiple signed languages rely on repetition (i.e., reduplication) to encode grammatical features such as switching verbs to nouns (in American Sign Language; 71), or the pluralization of certain nouns (in German Sign Language; 72; see SI for a related discussion about revealing communicative goals in vocal communication).

And then turn to presenting new predictions that our account makes in this domain:

At the same time, this does not imply that sign languages lack the need to reveal that movements are communicative. Signs may include intrinsic flexibility that allow the introduction of ad-hoc movements that help reveal communicativeness when necessary. Specifically, signs are thought to consist of three building blocks that vary in flexibility: handshape (which is highly conventionalized and least flexible), movement, and location (which is least conventionalized and most flexible, allowing signs to sustain their meaning, independent of the position in space where they are produced; 73-75). The degrees of freedom in a sign's movement and location have been hypothesized to allow signs to physically depict aspects of a scene or interaction (e.g., 74). However, it is also possible that signers use this flexibility to reveal that an action is communicative. When signers encounter ad-hoc environmental constraints that may make it harder for their sign to be recognized as communicative, they could modify the sign dimensions that are least conventionalized to increase the sign's rarity. This may be especially useful in the context of iconic or pantomimic signs, which may resemble world-directed action; changing their location or movement away from objects in the environment that could confuse observers may enable signers to maintain the representational nature of the signs while also helping observers infer their communicative goal (e.g., 55-56). This view predicts that, within sign languages, observers might have a stronger expectation that position and movement are shaped to reveal non-world-directedness, relative to handshape.

Taken together, this view predicts a tradeoff between linguistic pressures and inferential demands: The expectation for communicative action to reveal its purpose should be stronger for movements that are less immersed in complex linguistic systems and movements that can be produced at any moment (rather than exclusively within complete grammatical utterances). If this is correct, then the expectation that communicative action will reveal its goal should be strongest for emblems—a type of conventional gesture that can be produced and understood in isolation, and at any point independent of the context (e.g., waving hello, or a thumbs up; 21-22). This is a prediction that we hope to test in future work.

We also discuss the developmental angle of how infants identify communicative signals in a more detailed discussion in a new SI section titled “Vocal Communication and the Pressure to Reveal Communicative Goals” :

Speech—much like signing—most often occurs in extended communicative interactions. This means that, as soon as a sound is recognized as speech, the listener can continue to expect the following sounds to also be speech until an utterance has been completed. Moreover, speech occurs in contexts where agents engage in turn-taking interactions with ostensive cues that are salient to even infants [7-10]. This context likely helps listeners assume that the sounds are communicative and may reduce the burden on the sound itself to reveal its communicative goal. Indeed, the effects of a communicative context on the interpretation of sound are so strong that it can even lead infants to treat highly artificial non-biological sounds as communicative. In one recent study, infants learned abstract patterns from sine-wave tones (a phenomenon previously proposed to be specific speech-based communication) when the tones were dubbed over a communicative interaction, but not when the same tones were presented in a noncommunicative context [11].

Throughout, we have made an effort to connect our work to the literature on ASL, citing the papers that we believed were most relevant (including, among many others, the Novack et al., 2016; Wakefield et al., 2018; and Ferguson et al., 2016 papers that the reviewer helpfully brought to our attention). This revision does not currently cite Rain Bosworth’s work, only because we struggled to identify the best way to connect it to our work (although we found it highly interesting and rewarding to read). However, we would be happy to integrate this work if the reviewer believes this is important.

Finally, we wish to flag that our revised discussion states that how novel gestures balance signaling meaning and signaling their communicative purpose is largely an open question (shown at the end of the first quoted paragraph in this response). This was based on our reading of the literature, but if the reviewer feels that we have in any way mischaracterized the status of this question or that other papers warrant inclusion here, we would be happy to update this section accordingly.

1.5. I was somewhat surprised by the finding (and the theoretical argument) that participants did not find it difficult to construe the rare, repetitive actions as world-directed. The logic of this theoretical prediction should be unpacked. E.g. perhaps the thinking is: if people could not construe these actions as world-directed, perhaps they could be using a simpler heuristic rather than a rational inference? However it seems to me that this finding is not an obvious prediction of the Bayesian model. In particular, the Bayesian inference people are making can be stated as: Which goal is the best explanation for this behavior? A physical-world-related explanation? Or communication? So, in order to come to the conclusion that communication is the best explanation, one must think that the actions are less well-explained by (or less consistent with) physical-world-related goals. In the extreme, people thus should find it harder to explain these actions through physical-world-related goals — or, if not harder, at least they should find these explanations less compelling or convincing. In any case - it seems necessary to unpack the

prediction regarding how hard or easy it should be to construe rare/repetitive actions as having world-directed goals, and how it falls out of the model.

We thank the reviewer for highlighting this point of confusion. The reviewer is correct that our concern is that participants could use a simple heuristic along the lines of “if I can’t think of a world-directed goal, I guess I will say that the movement is communicative” without actually believing that the movement is communicative. Our previous manuscript did not make this clear that this was a methodological concern that does not fall out of the Bayesian model.

We have revised the manuscript to explain the motivation and logic of the Explanation Control more clearly. On pages 9-11 of the manuscript we now write:

Our results so far show that people judge rare and repetitive paths as more likely to be communicative. There are at least two possible mechanisms behind these judgments, both of which are consistent with our account. A first possibility is that people can flexibly interpret the movement as world-directed or as communicative, but find a communicative interpretation to be more suitable for rare and repetitive movements (based on the relative likelihood of observing the behavior under each interpretation). A second possibility is that people are simply unable to interpret rare and repetitive movements as world-directed, and therefore endorse a communicative interpretation. While both processes are consistent with our account, the second one raises a methodological concern: is it possible that participants did not believe that the movements looked communicative, and their responses reflected only a confidence that the movements could not be world-directed?

To explore this potential confound, we tested participants’ ability to invoke non-communicative explanations for all movements used in Study 1. We reasoned that if participants can easily invoke non-communicative goals as explanations, independent of path rarity and repetition, then the results of our study would suggest that participants were actively endorsing a communicative interpretation (rather than defaulting to it because they were at a loss about what else to do). Participants in the Explanation Control watched the same videos from Study 1, but were given a context where communication was unlikely (see Methods). Participants were asked to infer the agent’s goal and rate how difficult it was to conceive of this explanation. If Study 1 judgments reflected an inability to think of non-communicative goals, then path rarity should correlate with difficulty of explanation (revealing that participants rated rare paths as communicative only because it was difficult to think of an alternative interpretation, replicating the correlation found in Study 1). Instead, we found no relation between rarity and difficulty of explanation (Explanation Control: $r=0.10$, $CI_{95\%}=-0.31-0.56$; replication: $r=0.38$, $CI_{95\%}=0.14-0.83$; Fig. 2d), suggesting that people can easily conceive of non-communicative goals for complex movements. Thus, responses in Studies 1-3 likely reflect participants’ belief that a communicative goal was a better explanation, rather than defaulting to this answer due to an inability to think of any possible alternative (see SI for additional analyses revealing this null effect was not due to task misunderstanding, and Studies 6-7 for additional evidence that rare and repetitive movements are actively seen as communicative).

1.6. Some of the work cited showed that when movements are inefficient as a means to any object or location, including (but not limited to) repetitive movements, people often infer that the goal was to produce the particular movements, for their own sake (rather than for any physical-world-directed goal - Schachner & Carey 2013). How should the reader think about the relationship of those findings to these? It seems like the current model would predict that the stimuli in this older work should be seen as communicative.

Thank you for this comment. We agree that, under our theory, the movements used by Schachner and Carey (2013; henceforth S&C) ought to be seen as communicative. We believe that the S&C stimuli did not elicit communicative inferences because the movements were generated in a context that ruled out this possibility. This happened for two reasons.

First, communicative action is typically accompanied by ostensive cues like attention towards the recipient. Under our account, ostensive cues help detect that an agent *intends* to communicate, and an efficiency-based analysis enables us to infer which aspects of a movement are communicative (reviewed in paragraph 2 in our Introduction). The agents in S&C performed the actions while looking towards the sky or to its side:

Therefore, while the movements might appear to be communicative, the agent's attention towards the sky reduces the possibility that the agent intended to communicate.

The second reason why the videos in S&C might suppress a communicative interpretation is because all videos show a single agent, from a far-enough viewpoint that viewers might reasonably infer that the agent was alone. Combined, the apparent absence of an agent, and the actor's attention towards the sky, may have made participants less likely to interpret the movement as communicative and infer that the goal of the movement was the movement itself.

To test if our interpretation is correct, we have conducted an additional experiment using altered versions of S&C Study 1, with two key changes. First, the videos show the agent from behind so that their eye gaze is not visible (enabling us to control for ostensive cues). Second, we included an observer in the scene. In line with the original study, we had participants complete an

objects-present and an *objects-absent* condition (tested across participants), and answer the question “In the video, what was Tim’s intention?”.

As the figure below shows, our experiment replicated S&C’s objects-present condition, finding that participants infer external goals from the movement (75% of participants in our study). As predicted, our objects-absent condition diverged from the findings of S&C. Although we used the exact same movements that S&C did, our variant led 42% of participants to infer a communicative goal, a quantity comparable to the percentage of participants in the S&C study that inferred that the goal was to produce the movement itself (49% of participants). See figures below for a comparison between our data and the data from S&C.

These findings show that, as predicted by our theory, the movements used in S&C are detected as communicative when they reveal the absence of a physical goal. At the same time, they also provide additional support for S&C's theory, showing that movements are seen as having their own goal when no other explanation (including communication action) is available.

Additionally (as the reviewer comments below), goals are fundamentally hierarchical (see response R1.7 below for changes in light of this comment) and communicative goals can also be viewed as a type of movement-based goal. Therefore, the addition of an agent to the scene may have increased participants' tendency to describe the goals according to a higher-level communicative goal, rather than the lower-level movement-based goal.

Our revised manuscript now includes this experiment in the SI in a section titled "Supplementary Study based on Schachner & Carey (2013) Study 1", and our main text directs readers to it in the Discussion (Page 22). We hope that this supplementary study is of interest to readers actively researching in this area since it opens interesting questions about when we articulate higher vs. lower level goal explanations for movements.

1.7. This points to one of the most complex issues of action representations: They are hierarchical. That is, a single action is seen as having multiple goals, at different levels of abstraction. Thus, a single reaching action may have the goal of "grasping a handle" and also the goal of "drinking coffee" and also the goal of "becoming more awake" and also the goal of "finishing this manuscript". One possibility is that the actions under investigation here, as well as

those in the work by Schachner & Carey, are an overlapping set. In other words, an inefficient, repetitive action may be seen as having the goal of performing those particular repetitive movements (and thus what they term a movement-based goal) - but at the same time, at a more abstract level, have the goal of getting an observer to recognize that they are trying to communicate. This is one possible interpretation — there are other very reasonable interpretations as well. However, it would be useful to provide an account that more deeply integrates these findings. (This integration, along with the below suggestions for links to other accounts, would serve to flesh out the claim that “not all deviations from world-directed actions are communicative” (page 22 1st paragraph), a line with important implications which should be more fully developed.)

Thank you for this comment. We agree that goals are best understood hierarchically, and our revised manuscript addresses this concern in several ways.

First, our manuscript is now clearer with respect to how conventional action and communicative action overlap (in line with Reviewer’s point 1 above; R1.2), and we believe this positions our work more clearly under a framework where action categories are not mutually exclusive. Second, our revised manuscript now also presents an extended discussion of our work in the context of sign language, which includes a discussion of how signs must be represented at multiple levels (as an action intended to communicate, and as a representation), in line with Reviewer 3’s comment below (see R3.2 for details).

Finally, our discussion directly addresses this nuance in the context of our work:

Our work treated world-directed and communicative goals as mutually exclusive, but this is often not the case. Goals have a fundamental hierarchical structure and they can be explained at varying levels of abstraction. For example, a single action performed in front of an observer may be conceptualized as “moving a hammer up and down,” “hammering a nail,” and “showing how to properly use a hammer.” From this standpoint, our studies not only show that people see rare and repetitive movements as communicative, but that the features that trigger this inference may also lead people to favor this level of explanation over lower-level ones (particularly in our open-ended response task; Study 7). Consistent with this, related work has found that, when agents produce rare or repetitive movements in a context where both communicative and world-directed goals are implausible, people default to the lowest-level explanation, judging that the agent’s goal was to simply produce the movements themselves (7). In these cases, however, a communicative-level explanation is recovered when an observer is present (see SI for an additional study replicating the initial effect and an extension showing when communicative-level explanations are recovered).

1.7 I suggest that the terminology used should be modified, to help the reader hone in on the crux of the issues. Particularly, with regard to the terms “world directed” and “rare”.

The term “world-directed”. This refers particularly to interactions with the physical world (as in the example provided: reaching for and manipulating objects). This is contrasted with

communication. While this conceptual distinction is fine and useful to make, the term world-directed does not do justice to this distinction. After all, other people are part of the world as well — the social world. Communication is an act meant to have an impact on other people (and thus the world). Thus the distinction in the paper may be better labeled as directed toward the physical world versus the social world (changing to physical-world directed vs. communication would be enough to make this clearer, perhaps).

We agree with the reviewer that “world-directed” does not capture the nuance of this distinction. To address this concern, we initially replaced all mentions of “world-directed” with “physical-world directed.” However, when proof-reading the manuscript, we found that the new term made the sentences harder to read, increasing the burden on readers.

Unfortunately, we were unable to come up with a simple term that fully captured this distinction while also being sufficiently concise. Therefore, to address this concern, the revised manuscript presents a clear definition of our term “world-directed”, and we remind readers of the meaning of this term the first time that it is mentioned in the introduction, in the results section, and in the general discussion.

In proof-reading the revised version, we believe that the multiple reminders of our use of world-directed helps solve this concern. However, if these changes do not fully alleviate the concern, we would be happy to update every instance of “world-directed” for “physical-world-directed”, or a different term if the reviewer believes this would be beneficial.

1.8. The paper uses the term “rare”, with the claim that rare actions are seen as more likely communicative. Rare implies unusual, or not typically produced. However what is termed rarity is actually quantified in the paper as inefficiency — “how much each movement diverged from the shortest path”. Why say rarity when what is meant is inefficiency toward physical-world-directed goals? In my view using the term “inefficiency” rather than rarity would help to clarify the relationship of this work to prior work in action understanding, in a useful way. For observers, inefficiency toward other goals provides powerful evidence for exactly what the Bayesian model in the paper instantiates — that the actions are not well-explained by the whole class of physical-world-directed goals.

Thank you for this comment. We agree that our previous manuscript was confusing in this respect and we thank the reviewer for bringing this to our attention. Our goal is to propose that actions that efficiently reveal non-world-directedness are seen as communicative. While this idea is deeply related to inefficiency, these two concepts are not always equivalent, as there are many types of inefficiency that fail to make a movement efficiently reveal it is not world-directed. For instance, an agent zig-zagging towards a goal would be acting inefficiently, but this behavior would not be considered communicative under our account.

We believe that our manuscript was confusing in this respect for two reasons. First, we failed to make this distinction early and clearly. Our revised manuscript now articulates this distinction in the introduction, and returns to its importance in the discussion. Please see response R2.2

(from Reviewer 2, who suggested that this distinction be made early in the manuscript) for a full list of the changes that we made to remove this point of confusion.

Second, the first experiment seeking initial evidence for our account (Study 1) used inefficiency as a pre-registered first-pass approximation for rarity. The fact that rarity and inefficiency correlate but are not identical was the underlying motivation for using a computational model to analyze the data. This is because our computational model implements a more complex notion of rarity; the model can successfully infer world-directed goals even when agents are inefficient, and it therefore does not treat all inefficiency as communicative. Similarly, all of our studies using body movements quantify perceived rarity in terms of unexpectedness when pursuing world-directed goals rather than as inefficiency.

Beyond marking more clearly how our concept of rarity differs from efficiency, our revised manuscript now clarifies how our first analysis equates rarity to inefficiency as a first-pass approximation, and then clearly uses this limitation to motivate our use of a computational model. Specifically, the caption where we present Study 1 now reads:

Figure 2. Results from Studies 1-3 and the Explanation Control, combining original studies and their pre-registered replications. a) Average communicativeness judgments (y-axis) as a function of the path's rarity (x-axis) in Study 1, operationalized here as deviation from the shortest path (see Methods). b) Average communicativeness judgments (y-axis) as a function of number of path repetitions (x-axis). Vertical lines show bootstrapped 95% confidence intervals. c) Study 3 results as a function of rarity and condition. Each point represents a path with the average communicativeness in the bordered condition (x-axis) and the unbordered condition (y-axis). Smaller circles represent rarer paths. Points above the diagonal line indicate a higher communicativeness rating in the unbordered condition relative to the bordered condition. The difference across conditions was larger for paths with higher rarity ($\beta_{\text{rarity:condition}}=1.14$, $p=.002$; replication: $\beta_{\text{rarity:condition}}=0.92$, $p=.015$), further suggesting that these paths were no longer seen as communicative because environmental constraints in the bordered condition removed the paths' perceived rarity. d) Average reported difficulty of generating a world-directed explanation as a function of the path's rarity (x-axis) in our Explanation Control. If Study 1 judgments were driven by a pure inability to consider world-directed explanations, then the difficulty of explanation should show a strong positive correlation with rarity, but this was not the case.

Our introduction to the modeling section now discusses this issue:

Our first study approximated rarity by quantifying deviations from the shortest path, but not all inefficiencies are rare (see Introduction). To test a more nuanced view of rarity, we took computational models of goal inference (10,12-13,27) and modified them to compute the likelihood that movements might appear communicative. Critically, these models can naturally distinguish between inefficiencies that are consistent with world-directed behavior (e.g., zig-zagging towards a goal) and inefficiencies that cannot be explained by any world-directed goal.

With the caption from our modeling results reading:

Figure 3. Results from a computational model that determines communicativeness when the movement reveals that it lacks a world-directed goal. Critically, this model has a more nuanced concept of rarity, as it can infer world-directed goals for inefficient movements (such as an agent zig-zagging towards an object). a) Model predictions showing the belief that each path from Study 1 was world-directed as a function of time. Each line represents one of the 23 videos from Study 1 (partially occluded due to over-plotting), with frame number on the x-axis and model prediction on the y-axis. Lower model predictions indicate that the movement looked less world-directed (in log-space; see Eq. 1). Each line's shading indicates the average communicativeness rating received for that video in Study 1. b) Final model predictions (x-axis) against participant judgments (y-axis) from Study 1. To make scales comparable, we transformed model ratings into communicative inferences through a linear regression predicting participant judgments based on the model's final output (see SI for details). Each point represents a path's model prediction (x-axis; negative log of probability of a world-directed goal given the full trajectory—equivalent to the predictions on frame 12 of panel a)—against average communicativeness ratings from Study 1 (y-axis). c) Disagreement between model predictions and participant judgments as a function of the movement class (see Fig. 1 for movement class labels). Positive numbers indicate that the model saw the movement as more communicative than participants and negative numbers indicate that the model saw the movement as less communicative than participants.

Minor comments:

1.9. Page 18 paragraph 1 last line: is this percentage the percent reporting another person is present, or absent? This is currently ambiguous based on the surrounding text.

We thank the reviewer for pointing this out. We have clarified that the statistic refers to the percentage of trials in which the participants responded that the demonstrator was alone.

1.10. P.11 Paragraph 3 - here the prediction of the account is stated, but in this context this prediction needs to be unpacked. I suggest explaining or briefly stating how this prediction falls out of the model, at this point in the paper.

We appreciate this feedback. We realized this paragraph was confusing because we meant to highlight a component that does not fall out of our model. We have adjusted our wording to make this clearer:

Critically, however, these predictions only considered the final belief that a movement was not world-directed, without taking into account how quickly this occurred. Under our proposal, movements that reveal non-world-directedness quickly should be seen as more communicative relative to paths that reveal non-world-directedness slowly.

1.11. p. 7 line 8, need a semi-colon before Fig.2a

1.12. p. 8 line 7, Fig.1d should be Fig. 2d

We apologize for these errors and have fixed them in the manuscript.

Reviewer #2 (Remarks to the Author):

2.1. This paper reports a series of elegant experiments that advance our understanding of communication. Overall, I am enthusiastic about the paper. Both of the novel tasks designed to study expectations about communication are innovative and well-controlled, providing a substantial empirical contribution. The paper is also making a number of useful conceptual contributions to the literature and will likely have impact on future studies of communication and action understanding. However, there are several issues, including questions concerning the generalizability of the results, that I would like to see addressed in a revision.

We appreciate the reviewer's kind words and thoughtful feedback!

Major points:

2.2.. "Rarity" should be defined early on in the paper and it should be made clear how the concept of rarity differs from efficiency.

Thank you for bringing this to our attention. Our revised introduction now presents an extended explanation of rarity. On page 4-5 of the manuscript we write (key text highlighted only here and not in main text):

Our model-based analysis shows that communicative action should be shaped so that, from the onset, it is unlikely to be produced while pursuing world-directed goals. In other words, communicative movements ought to be *rare* under the distribution of movements that people produce when acting on the physical world (consider, for instance, a body position that is unlikely to ever be generated when interacting with the world, such as a thumbs up). Although rarity often refers to the statistical frequency of an action, here we use it as a shorthand to refer to movements that are uncommon when agents pursue world-directed goals. Critically, rarity is related to, but different from inefficiency: some deviations from efficient world-directed action (e.g., deviations due to errors, pursuing subgoals, circumventing hidden obstacles, or movement idiosyncrasies) are inefficient, but still occur when agents pursue world-directed goals (11,30-31) and are therefore not rare under our definition. Our analysis also revealed one particularly important type of rarity: repetition. That is, agents can make movements rare simply by repeating them, without changing the world. If people expect communicative actions to efficiently reveal that they are not world-directed, then rare and repetitive movements ought to be seen as communicative.

Combined with the changes that we made to address point R2.4 below, our Discussion touches again on the distinction between rarity and inefficiency when discussing limitations

Our studies focused on rarity (and repetition, as a mechanism for making movements rare) as a way to efficiently reveal that a movement is not world-directed. This category, however, was derived by considering a space of simple goals similar to those that even

infants understand (i.e., agents moving efficiently in space towards target locations; 6,8,28). It is likely that additional features would emerge under a richer model of world-directed action (particularly models that include the inefficiencies associated with world-directed goals, such as those introduced by incompetence or information seeking). Indeed, a movement's velocity, size, and punctuality also influence the recognition of communicative action (43-46). These features might further support the recognition of communicative action precisely because they efficiently reveal that the movement is not world-directed. Similarly, research in cognitive neuroscience has found that objects that are typically moved towards the body (e.g., combing one's hair) are represented differently from objects that are typically moved away from the body (e.g., hammering a nail; 47). The rare and repetitive movements used in Studies 4-7 were all deviations from body-directed actions (e.g., wiping off hands, moving hair), and it is possible that additional features will emerge when communicative actions need to disambiguate themselves from goals directed towards inanimate objects apart from the body. Thus, our work provides broad support for the idea that communicative action is expected to efficiently reveal a lack of world-directed goals, but leaves open the question of how people expect this principle to shape the spatiotemporal structure of action in more complex events.

And we come back to this again in the context of research discussing how inefficient movement can support world-directed interpretations of action:

While our work focused on explicit communicative action, our theoretical framework shares similarities with work in several adjacent fields. First, research in sensorimotor communication has found that people pursuing world-directed goals spontaneously introduce inefficient movements to help observers quickly identify which goal they are pursuing (e.g., curving a hand movement more than necessary to make its direction clear; 44-45,48-49). This work shows further support for the idea that not all inefficiency is communicative, as some inefficiencies serve the purpose of further establishing a world-directed goal. However, the world-directed inferences that people can make from small inefficiencies, and the communicative inferences that people make from action that reveal non-world-directedness may arise from the same inferential system, where observers analyze action by considering how well an inefficiency is designed to reveal that an agent does not intend to act on the world.

While our work focused on explicit communicative action, our theoretical framework shares similarities with work in several adjacent fields. First, research in sensorimotor communication has found that people pursuing world-directed goals spontaneously introduce inefficient movements to help observers quickly identify which goal they are pursuing (e.g., curving a hand movement more than necessary to make its direction clear; 44-45,48-49). This work shows further support for the idea that not all inefficiency is communicative, as some inefficiencies serve the purpose of further establishing a world-directed goal. However, the world-directed inferences that people make from small inefficiencies, and the communicative inferences that people make from rare and repetitive actions may arise from the same inferential system, where observers analyze movements to decide if the inefficiency is designed to accentuate an intention to act on the world, or an intention to convey a message.

We hope that, together, these changes now support a clearer presentation of rarity.

2.3. As far as I understand, participants in all studies were made to believe that if there was any communication, it was based on a previously established, conventionalized, communication system. So, when participants performed the task of judging whether observed actions were or were not communicative, they were judging whether observed actions were part of a conventionalized communication system, rather than judging whether someone was trying to communicate impromptu in the absence of an established system. If my understanding is correct, then I would find it important that the authors make this point more explicit in the paper and discuss whether they would expect the results to generalize to a version of the experiments that does not mention a previously established code in the instructions. I find this important because people might have different assumptions about established communicative actions versus non-conventionalized communicative actions.

Thank you for this comment and we agree with the reviewer. Our revised manuscript now makes this explicit. First, our introduction makes the scope of our studies clearer:

Throughout, we focused on one-shot events where people must detect conventional communicative action (which was novel to participants) with minimal linguistic or environmental cues. This enabled us to test our hypothesis while controlling for factors that might further affect how people reason about communicative action, including expectations about how form maps to meaning (e.g., iconicity), and how individual gestures fit into complete linguistic systems. Equipped with our results, we return to these points in the discussion and present the implications of our framework for gestural communication in more complex settings.

And we remind readers of this point when we begin describing our studies and results:

In Studies 1 and 2, participants watched videos of a point moving in a two-dimensional plane, and their task was to infer whether the movements were conventional communicative actions intended for an observer with a bird's eye view (Fig. 1a). As predicted, we found a strong correlation between rarity (how unlikely it is that the movement would be produced if pursuing a world-directed goal) and communicative inferences (Study 1: $r=0.80$, $CI_{95\%}=0.67-1$; replication: $r=0.87$, $CI_{95\%}=0.79-1$; Fig. 2a). Moreover, increasing the number of repetitions in the movements while keeping all other features constant led to a significant increase in people's communicativeness ratings (Study 2: $\beta_{\text{repetitions}}=1.40$, $p<.001$; replication: $\beta_{\text{repetitions}}=1.43$, $p<.001$; Fig. 2b).

Finally, our discussion then returns to this point to discuss how our proposal applies to non-conventional communicative action:

Our work also opens two further questions regarding humans' inferences about communicative goals. First, participants in our studies were told that all communicative

actions being produced were conventional (although unfamiliar to the participant). Do people hold the same expectations about novel or ad-hoc communicative action? In these situations, people's expectation that communicative action should reveal non-world-directedness might be even stronger, as the lack of a convention could increase communicators' pressure to ensure that a novel gesture is recognized as such. At the same time, however, novel and ad-hoc gestures must go beyond revealing their purpose (i.e., to communicate) and they must also reveal their meaning (i.e., what the gesture represents), due to the lack of a convention. This suggests that observers might expect novel communicative action to simultaneously reveal its communicative purpose and its meaning. Related research suggests that people's inferences about the meaning of novel gestures are commonly grounded in an expectation for pantomime and iconicity (e.g., if a movement *looks* like hammering, then it probably represents hammering; 54-55). This appears to create a challenge for gesturers, as their movements must often resemble world-directed action while also revealing that they are not world-directed action. Recent research into gesture provides one potential solution: Gesturers might solve this problem by relying on the context to ensure that the movement cannot be incorrectly perceived as world-directed (e.g., reaching to touch an object but stopping before actually touching it, ensuring that observers recognize the actions did not change the state of the world; 56-57). To our knowledge, however, how people balance the pressure to reveal communicative action and the pressure to make its meaning decodable remains largely an open question.

As our Discussion states above, to our knowledge, how novel gestures balance signaling meaning and signaling their communicative purpose is largely an open question. We cited what we saw as the most relevant papers, but if the reviewer feels that we have in any way mischaracterized the status of this question or that other papers warrant inclusion here, we would be happy to update this section accordingly.

2.4. The authors seem to assume that instrumental goals (or world-directed goals in their terminology) are performed most efficiently without repetition and by minimizing distances to be covered. However, there are instrumental goals, like searching for something or exploring a space, that involve repetitions and necessary deviations from the shortest path. For example, if participants in Study 1-3 had been told that the anthropologist is either communicating or searching for her glasses that she has lost, the results might be quite different. It seems important to acknowledge that the distinction between instrumental and communicative actions is not always as clear-cut as the present studies suggest, and to discuss whether the results would generalize to situations where instrumental actions include exploration or searching behavior.

Thank you for pointing out this nuance. We agree that there are many behaviors that involve inefficient movement but are not communicative. Under our account, these inefficiencies would not be communicative because they do not efficiently reveal non-instrumentality¹. This is

¹ In our manuscript we use the term 'world-directed' in place of 'instrumental' so that the work would be easier to read for an audience that is less familiar with research on goal inference. In this section of the response we adopt the term 'instrumental' to maintain the letter consistent with the reviewer's terminology.

because, as the reviewer correctly notes, such inefficiencies are consistent with an instrumental interpretation.

We also agree that many instrumental inefficiencies involve repetition (such as knocking on a door). Since repetition is not a context-independent cue to communicativeness, we believe that this nuance is consistent with our account: we propose that recognizing communicative action is a context-sensitive inferential process (rather than a cue-recognition system). From this standpoint, repetition (and rarity) trigger communicative inferences in our study because they efficiently reveal non-instrumentality in many contexts, but such interpretation can be overridden in situations where an instrumental interpretation is justified.

We have made several major changes to our manuscript to address this point. First, our revised introduction raises this point when we introduce our key predictions. In our introduction, when introducing our predictions, we write (key change underlined):

Our model-based analysis shows that communicative action should be shaped so that, from the onset, it is unlikely to be produced while pursuing world-directed goals. In other words, communicative movements ought to be *rare* under the distribution of movements that people produce when acting on the physical world (consider, for instance, a body position that is unlikely to ever be generated when interacting with the world, such as a thumbs up). Although rarity often refers to the statistical frequency of an action, here we use it as a shorthand to refer to movements that are uncommon when agents pursue world-directed goals. Critically, rarity is related to, but different from inefficiency: some deviations from efficient world-directed action (e.g., deviations due to errors, pursuing subgoals, circumventing hidden obstacles, or movement idiosyncrasies) are inefficient, but still occur when agents pursue world-directed goals (11,30-31) and are therefore not rare under our definition. Our analysis also revealed one particularly important type of rarity: repetition. That is, agents can make movements rare simply by repeating them, without changing the world. If people expect communicative actions to efficiently reveal that they are not world-directed, then rare and repetitive movements ought to be seen as communicative.

Our Discussion then raises the point that rarity and repetition specifically can also be associated with instrumental goals (key changes underlined):

Our work shows that the ability to recognize and interpret body movements as communicative is grounded in a context-sensitive inferential process, where observers expect communicative action to efficiently reveal that it is not meant to change the physical world (i.e., not world-directed). Critically, people's communicative inferences were not determined purely by movements' superficial structure and they were sensitive to whether contextual information supported a world-directed interpretation. These results are consistent with everyday situations where rarity and repetition are often present in communicative action (e.g., waving, nodding, and winking), but are treated as world-directed when the context supports this interpretation (e.g., scratching is interpreted as world-directed despite its repetition). Indeed, work in ethnography has noted that people often 'hide' offensive gestures by producing them in a context that

supports a world-directed interpretation, giving them plausible deniability (e.g., someone using a middle finger to scratch their eye: 40-42). Altogether, this suggests that rarity and repetition trigger communicative inferences because they efficiently reveal non-world-directedness in a wide range of contexts, but observers do not treat these movement features as superficial cues that mark communicativeness in a context-independent manner.

And we highlight that we expect our theory to predict more complex signatures (that go beyond simple inefficiency) under more complex models of instrumental action:

Our studies focused on rarity (and repetition, as a mechanism for making movements rare) as a way to efficiently reveal that a movement is not world-directed. This category, however, was derived by considering a space of simple goals similar to those that even infants understand (i.e., agents moving efficiently in space towards target locations; 6,8,28). It is likely that additional features would emerge under a richer model of world-directed action (particularly models that include the inefficiencies associated with world-directed goals, such as those introduced by incompetence or information seeking). Indeed, a movement's velocity, size, and punctuality also influence the recognition of communicative action (43-46). These features might further support the recognition of communicative action precisely because they efficiently reveal that the movement is not world-directed. Similarly, research in cognitive neuroscience has found that objects that are typically moved towards the body (e.g., combing one's hair) are represented differently from objects that are typically moved away from the body (e.g., hammering a nail; 47). The rare and repetitive movements used in Studies 4-7 were all deviations from body-directed actions (e.g., wiping off hands, moving hair), and it is possible that additional features will emerge when communicative actions need to disambiguate themselves from goals directed towards inanimate objects apart from the body. Thus, our work provides broad support for the idea that communicative action is expected to efficiently reveal a lack of world-directed goals, but leaves open the question of how people expect this principle to shape the spatiotemporal structure of action in more complex events.

2.5. The explanation control results are not fully convincing, as the judgments are consistently around 4. (How) can the possibility be excluded that the question just did not make much sense to participants? To get at this, can the authors report the variability for each data point in Fig 2 d)?

We agree that the pattern of data could be consistent with participants not understanding the task. We have addressed this concern in two ways.

First, we reasoned that if the task or questions did not make sense to participants, then it would be unlikely that they would answer the open ended question appropriately, but not be able to report on their own experience of answering that question. To test for this, we coded all participants' written responses to check whether they had produced responses that were

consistent with the scenario presented in the cover story and that suggested they understood the task. Of the 1,308 responses collected in the Explanation Control and the Explanation Control replication, only 14 were coded as unreasonable or nonsensical (for example, one participant said that the anthropologist was flying in an airplane, which is inconsistent with the scenario presented). This information is now presented in our SI section on the Explanation Control Study under “Responses”.

Second, as suggested by the reviewer, we have directly analyzed variability behind each data point in Fig 2d. We interpreted the reviewer’s comment as suggesting that, if participants did not understand the question, their answers should be clustered around the midpoint (4 in our seven-point scale) with little variability (suggesting that participants did not know what to respond), or with the same variability across all trials. However, if participants understood the question, their responses should reflect the difficulty that they experienced when generating different world-directed explanations. Given that the explanations that participants produced were highly variable (even within the same stimuli type), we expected the distribution of responses to be highly variable as well across trials.

As the figure below shows, participants did not uniformly favor entering the midpoint (4) for all trials, and all paths did not have the same distribution of answers. Instead, each trial shows a different pattern of responses. For instance, in trial *ihep* (see figure below for explanation for trial names), participants showed a bimodal distribution, where roughly half of the participants found it easy to generate an explanation and half of the participants found it difficult. By contrast, in trial *abkj*, participants were equally likely to find the process of generating an explanation easy, average, or difficult. Altogether, this figure suggests that participant answers reflected how difficult they found it to generate explanations, rather than a poor understanding of the task.

Histograms of the participant ratings for each of the 23 stimuli showing the frequency (y-axis) of different difficulty ratings (x-axis). Each subplot label indicates the stimulus path (see figure below for an explanation of each label's meaning).

Explanation of naming convention in figure above. Each path in Studies 1-3 was made up of four primitive paths from a set of 16 possible primitives shown in the grid above (See Methods Study 1-3 section “Stimuli” for a detailed explanation of path creation). Each primitive was assigned a letter; thus each path can be represented as a sequence of four letters. *inin* and *ibla* are two examples of paths and their corresponding letter sequences.

Our revised SI on the Explanation Control now discusses this concern and how we assessed participants’ written responses (sub-section “Responses” within “Explanation Control” section).

2.6. Concerning the model predictions for Study 1-3, how does the model fit change when the two movement classes with the largest discrepancy between model and participants (B and D in Fig 3 c)) are taken out? Does this significantly improve the model fit?

Removing the stimuli from classes B and D does improve the model fit from $r = .77$ to $r = 0.9$. For your convenience, we have reproduced the figure showing the original correlation and the correlation without classes B and D below.

In order to test whether removing the six data points in classes B and D significantly improves the fit model beyond what one would expect from removing any six points, we conducted a permutation test with 10,000 samples. This enabled us to test whether the improved correlation could be explained simply by having less data in the correlation, which can sometimes lead to inflated effect sizes.

For each sample, we computed one correlation between the model predictions and the participant responses based on randomly removing six data points. This was repeated 10,000 times to test whether the improved correlation between the model and participant data resulting from removing the six data points in classes B and D was significantly different from the distribution of the correlations based on random removal of datapoints. We found that correlation resulting from the removal of data points in classes B and D was significantly different from the distribution of correlations based on random removal ($r = 0.90, p < 0.001$).

We now note this in the manuscript and include the analysis in our SI section titled “Supplementary Model Analysis.”

A density plot showing the distribution of correlations between model predictions and participant judgments after removing six data points. The dashed vertical line represents the observed correlation coefficient after removing movements in classes B and D ($r = 0.90$).

2.7. In Studies 4-7, the "world-directed" actions could also be described as "body-directed". There is research in cognitive neuroscience suggesting that actions performed on the body/tools used on the body/words for actions and tools used on the body are represented in a different way than actions/tools/words for actions and tools use to act on the outside world, so it might be worth discussing the status of body-directed actions compared to actions performed on the environment.

We thank the reviewer for bringing this line of research to our attention. We now discuss how this distinction could impact the detection of communicative action:

Our studies focused on rarity (and repetition, as a mechanism for making movements rare) as a way to efficiently reveal that a movement is not world-directed. This category, however, was derived by considering a space of simple goals similar to those that even infants understand (i.e., agents moving efficiently in space towards target locations; 6,8,28). It is likely that additional features would emerge under a richer model of world-directed action (particularly models that include the inefficiencies associated with world-directed goals, such as those introduced by incompetence or information seeking). Indeed, a movement's velocity, size, and punctuality also influence the recognition of communicative action (43-46). These features might further support the recognition of communicative action precisely because they efficiently reveal that the movement is not world-directed. Similarly, research in cognitive neuroscience has found that objects that are typically moved towards the body (e.g., combing one's hair) are represented differently from objects that are typically moved away from the body (e.g., hammering a nail: 47). The rare and repetitive movements used in Studies 4-7 were all deviations from body-directed actions (e.g., wiping off hands, moving hair), and it is possible that additional features will emerge when communicative actions need to disambiguate themselves from goals directed towards inanimate objects apart from the body. Thus, our work provides broad support for the idea that communicative action is expected to efficiently reveal a lack of world-directed goals, but leaves open the question of how people expect this principle to shape the spatiotemporal structure of action in more complex events.

We had difficulty finding relevant studies beyond citation 47 in the manuscript (Rueschemeyer, S. A., Pfeiffer, C., & Bekkering, H. (2010). Body schematics: On the role of the body schema in embodied lexical-semantic representations. *Neuropsychologia*, 48(3), 774-781.)

We would be glad to incorporate further literature if the reviewer is aware of additional studies that would be important to review here.

Minor:

2.8. There is a numbering problem in the list of references: Reference 20/21 list 4 rather than 2 references.

Thank you for bringing this to our attention. We have revised the reference list accordingly.

2.9. p. 18 last sentence is not clear ("Among the subset...")

We thank the reviewer for pointing out this wording issue. We have revised the sentence so that it now reads:

Among the subset of responses that were categorized as either world-directed or communicative (63% of responses from US participants and 92.7% of responses from Tsimane' participants), communicative explanations were most often produced in response to rare + repetitive movements (US: 80.77% of rare+repetitive video trials, $CI_{95\%}=74.73-86.26\%$; Tsimane': 61.80%, $CI_{95\%} = 51.69-71.91\%$; Fig. 7b; see SI for supporting mixed effects model results) while world-directed explanations were most often produced in response to basic movements (US: 92.86% of basic video trials, $CI_{95\%}=89.29-96.43\%$; Tsimane': 84.27%, $CI_{95\%}=76.40-91.01\%$).

Reviewer #3 (Remarks to the Author):

3.1. This paper presents a series of studies of movements showing that certain characteristics of movements (rarity operationalised as deviation from direct paths between points, and repetition) relate to movements being more likely to be classified as communicative. The work presents a thoroughly conducted set of studies, testing these hypotheses about movements in different ways (movements on a map, and hand gestures) and on different populations. It is an impressive quantity of work. The studies have the potential to be of broad interest in terms of highlighting aspects of gesture and action that suggest communicativity of the signal to another, and my sense is that they would generate substantial interest over studies of communicative systems.

We are grateful for the reviewer's thoughtful comments and suggestions for further expanding the scope of our work.

3.2. There are a few issues that need to be strengthened to maximise the impact of the work presented in this paper.

First, iconicity is a key issue for sign languages (and spoken languages) but it is relatively sidelined in the paper (there are some references to work on this on p.22). As these references (and others - see work for instance by Vinson, Vigliocco, Perniss) point to iconicity as being hugely important as a characteristic of sign language systems, this needs to be foregrounded more. In particular, the definition of communicative is often characterised as non-goal directed, and this limits the way in which communicative can be interpreted (by the participants in the studies, and by the readers of the article). I would like to see greater discussion of iconicity and how rarity and repetition might relate to iconicity in sign systems. Iconicity seems to fall somewhat between goal-directed and non-goal-directed. For instance, in BSL the sign for eat is close to a mime to eat (but repeated) <https://www.britishsignlanguage.com/bsl-dictionary/eat/> . Is

that goal-directed, or communicative, or both goal-directed and communicative? That is not yet clear. If the paper is dealing only with aspects of communication that are not iconic then that needs to be stated. If the paper is addressing iconic signs as well as non-iconic signs then that needs to be outlined more clearly how that is the case. The discussion on p.23 seems to suggest iconicity is another aspect of signs in addition to rarity/repetition cues, but I think it might be very difficult to isolate it from the fundamental nature of signs in human communication.

Thank you for bringing these points to our attention. For simplicity, we begin by discussing how we address the reviewer's broadest point (mentioned in the context of the BSL sign for eating): our work was unclear about the relationship between different types of actions, including goal-directed and communicative. Similar concerns were raised by Reviewer 1 (who, in R1.7, suggested our work could be more clearly tied with issues about how any action can be represented at multiple levels of abstraction) and Reviewer 2 (who, in point R2.3, suggested that our be clearer about how participants were made to believe that the communicative actions were also conventional, and that our discussion present an analysis of what happens in contexts where actions are communicative but not conventional). We have edited our manuscript throughout to avoid giving a false impression that movements can fall only in one category and that the way people conceptualize action spans multiple and overlapping levels of representation. More details about these changes can be found in response to R1.6, R1.7, and R2.3.

With respect to the reviewer's main point, we agree that iconicity is a crucial feature of gesture and an important area where there are interesting and deep connections with our framework, but that our empirical work did not include. Our original intention was to establish a main effect predicted by our account, while controlling for other features that might further affect reasoning about communicative action. Our manuscript was insufficiently clear in delineating the scope of our empirical work and its implications and we thank the reviewer for bringing this to our attention. We have made two changes to the manuscript to address this concern.

First, our revised introduction now explicitly states our starting point, and we foreshadow a more substantial discussion of iconicity in the Discussion:

Throughout, we focused on one-shot events where people must detect conventional communicative action (which was novel to participants) with minimal linguistic or environmental cues. This enabled us to test our hypothesis while controlling for factors that might further affect how people reason about communicative action, including expectations about how form maps to meaning (e.g., iconicity), and how individual gestures fit into complete linguistic systems. Equipped with our results, we return to these points in the discussion and present the implications of our framework for gestural communication in more complex settings.

Next, our discussion now includes a new extended presentation of how our framework applies to sign languages, including the role of iconicity and pantomime:

Here we focused on isolated communicative action, but communicative action can also be embedded in complete linguistic systems such as American Sign Language and Nicaraguan Sign Language (59). How does our framework apply to communicative action in language? In the context of sign languages, the demand to continuously detect communicative action may be drastically reduced, for three reasons. First, observers in a conversation in sign language will already expect actions to be communicative. This may enable observers to treat all movements as potentially communicative and focus on recognizing each movement's meaning. Second, observers' knowledge of the sign language can further constrain the expectations about where in the body communicative action will be produced (e.g., expecting mouth morphemes in ASL; 60), alleviating the problem of revealing which aspects of someone's body movements are communicative and which are not. Finally, observers' real-time processing of an unfolding utterance can further constrain signers' expectations about which word will follow (relying on semantic constraints; 61-62). This may allow observers to anticipate the forms of subsequent signs for quicker identification of communicative action.

The reduced expectation that every sign must reveal its communicative purpose might also provide increased flexibility in the shape that signs can take. This could provide critical degrees of freedom that are needed to produce a more expressive system where movement encodes phonological (63-64), morphological (65), and syntactic (59) structure. Similarly, the prevalence of iconicity and pantomime in sign languages (66-69; especially in emerging sign languages; 70) suggests that signs may be subject to pressures to link form and meaning, which may also affect the shape of signs without regard to their rarity. It is possible that because of both the pressure to create linguistic structure in movement and the reduced inferential burden of detecting communicative action, sign languages may reuse features such as rarity and repetition for different purposes. For example, multiple signed languages rely on repetition (i.e., reduplication) to encode grammatical features such as switching verbs to nouns (in American Sign Language; 71), or the pluralization of certain nouns (in German Sign Language; 72; see SI for a related discussion about revealing communicative goals in vocal communication).

At the same time, this does not imply that sign languages lack the need to reveal that movements are communicative. Signs may include intrinsic flexibility that allow the introduction of ad-hoc movements that help reveal communicativeness when necessary. Specifically, signs are thought to consist of three building blocks that vary in flexibility: handshape (which is highly conventionalized and least flexible), movement, and location (which is least conventionalized and most flexible, allowing signs to sustain their meaning, independent of the position in space where they are produced; 73-75). The degrees of freedom in a sign's movement and location have been hypothesized to allow signs to physically depict aspects of a scene or interaction (e.g., 74). However, it is also possible that signers use this flexibility to reveal that an action is communicative. When signers encounter ad-hoc environmental constraints that may make it harder for their sign to be recognized as communicative, they could modify the sign dimensions that are least conventionalized to increase the sign's rarity. This may be especially useful in the context of iconic or pantomimic signs, which may resemble world-directed action; changing their location or movement away from objects in the environment that could confuse observers may enable signers to maintain the representational nature of the signs while also helping observers infer their communicative goal (e.g., 55-56). This view predicts that, within sign languages, observers might have a stronger expectation that position and movement are shaped to reveal non-world-directedness, relative to handshape.

Taken together, this view predicts a tradeoff between linguistic pressures and inferential demands: The expectation for communicative action to reveal its purpose should be stronger for movements that are less immersed in complex linguistic systems and movements that can be produced at any moment (rather than exclusively within complete grammatical utterances). If this is correct, then the expectation that communicative action will reveal its goal should be strongest for emblems—a type of conventional gesture that can be produced and understood in isolation, and at any point independent of the context (e.g., waving hello, or a thumbs up; 21-22). This is a prediction that we hope to test in future work.

We hope that this new section in the Discussion better situates our work in the context of sign languages, and the role of iconicity and pantomime.

3.3. Second, the authors include an additional classification in Study 7 as "descriptive". In the main paper analyses, descriptive responses were omitted, but they comprise a large proportion of responses, particularly in the US sample (Supp mats p.21, show 32% of basic movements were classified as descriptive and 33.3% of rare and repetitive movements). These are stable proportions across the types of movements and so the comparison between world-directed and communicative across movement types is not necessarily unduly affected, but how to take them into account? It could be that "descriptive" is closer to what other studies intend by "iconicity" in signs. In this case, there may be some difficulties in characterising signs as descriptive but not communicative. To whom is the signer describing if not another person? This descriptive signification seems like a key part of definition of what it means to be a sign and so it is odd to distinguish it from other communicative goals. Again, the authors need a more careful treatment and justification for the distinction between communicative and descriptive, how participants will have interpreted that distinction, and how that relates to human communicative systems.

Thank you for bringing this to our attention. We believe that our manuscript was accidentally misleading. Our "Descriptive" category was not meant to suggest that the participants in the study indicated that the actor was describing something to another agent with their movements. Instead, these were responses that simply recounted the actions performed without referencing any higher-level intention or goal. For example, two descriptive responses were "The person puts their left hand under their right hand and makes a motion with their right hand" and "The person touches their cheek with one finger."

Our original writing was confusing in this respect and we have edited the paragraph to make this clear. The revised text now reads:

Finally, to obtain direct evidence for communicative explanations, we ran an open-ended explanation task (Study 7), where US and Tsimane' participants were asked to explain what the demonstrator was doing (using basic and rare + repetitive low punctuality movements). Participant responses were then categorized as "world-directed" (aimed at producing a change in the physical world), "communicative" (produced with the goal of sharing a message with an observer), "descriptive" (describing the low-level actions performed by the demonstrator), or "other" (a catch-all category) by two coders who were blind to the video behind each explanation.

Our revised SI now also includes additional details about the descriptive category during coding and all answers are available on our OSF repository.

3.4. Third, the study omits mention of speech as a system of communication. But as this is the primary form of communication between humans it needs to be linked to the content of the paper more closely, otherwise the paper risks being seen as addressing only non-oral communication, which then limits the broader implications of the work. Should we expect speech to be (maximally) distinct from non-communicative oral productions, should we expect it to contain repetitions, or will it not fall under the same principles? If not, why not? As an example of how speech and non-speech communication can be compared, Perlman et al. (2018) showed that speech and sign languages can both be considered in terms of iconicity and there are parallels in terms of where iconicity is found, for different grammatical categories for instance, but in the current study it is hard to translate (excuse the term) ideas contrasting goal-directedness with rarity in the speech domain (there might be some mileage in repetition - a hallmark of child-directed speech for instance (though within words (syllables or phonemes) it is rarer than one might expect by chance).

We thank the reviewer for this suggestion. We now include seven paragraphs discussing the relationship between our framework and speech as well as other forms of vocal communication. In order to do justice to the richness of this problem, we needed to write a lengthy discussion. Attempts to shorten it to fit within the main manuscript forced us to omit important details, so we opted to move it to the SI (titled “Vocal Communication and the Pressure to Reveal Communicative Goals”) and reference it in the manuscript for interested readers, but we would be happy to move it to the main text if that is preferred:

Our work focused on the recognition of physical communicative action, including an analysis of how our framework applies to sign languages. But of course, spoken languages are often people’s primary form of communication. Here, we consider whether pressures to reveal a communicative goal apply to speech-based communication as well.

Vocalizations can be broadly thought of as falling into three categories: non-communicative non-speech vocalizations (e.g., coughing, yawning), communicative non-speech vocalizations (e.g., gasps, disgust vocalizations, but also vocalizations that are typically non-communicative, but can be used with a communicative intent, such as coughing to get someone’s attention, or yawning to intentionally communicate boredom), and speech (e.g., words and sentences). Therefore, if we apply our framework to the domain of vocal communication, then this would generate the hypothesis that people expect communicative vocalizations to reveal that they are not non-communicative vocalizations. We begin by considering this hypothesis in the case of speech, followed by the case of communicative non-speech vocalizations.

Identifying speech. There are two broad reasons why people may not need to expect speech to disambiguate itself from non-speech vocalizations.

The first reason is parallel to the reason why sign languages may not require every sign to disambiguate itself from world-directed goals (see Discussion in main text): Speech—much like signing—most often occurs in extended communicative interactions. This means that, as soon as a sound is recognized as speech, the listener can continue to expect the following sounds to also

be speech until an utterance has been completed. Moreover, speech occurs in contexts where agents engage in turn-taking interactions with ostensive cues that are salient to even infants [7-10]. This context likely helps listeners assume that the sounds are communicative and may reduce the burden on the sound itself to reveal its communicative goal. Indeed, the effects of a communicative context on the interpretation of sound are so strong that it can even lead infants to treat highly artificial non-biological sounds as communicative. In one recent study, infants learned abstract patterns from sine-wave tones (a phenomenon previously proposed to be specific speech-based communication) when the tones were dubbed over a communicative interaction, but not when the same tones were presented in a noncommunicative context [11].

The second reason why people may not need to expect speech to disambiguate itself from non-speech vocalizations is because this disambiguation might happen at a perceptual, rather than cognitive, level. Speech exhibits highly efficient neural encoding such that the greatest amount of auditory information is captured with the lowest neural effort [see 12 for review]. Similarly, infants prefer to listen to speech over non-speech stimuli (both when contrasted with acoustically similar artificial sounds and non-speech vocalizations such as gasps or disgust sounds) even when critical features of the stimuli are held constant [13,14]. As such, it has long been argued that speech has evolved a “special” status in comparison to other types of auditory input [15,16].

At an even broader level, it is possible that speech evolved as a primary medium of communication because those sounds were not historically used to perform non-communicative world-directed actions. Therefore this would make those sounds an ideal channel where listeners do not face high-levels of ambiguity when speech is produced. Note however, that there is significant debate about how and why the modality of linguistic communication evolved as it did [see 17 for review].

Identifying communicative vocalizations outside speech. Beyond speech, people often communicate through non-speech vocalizations, such as yawning or coughing. Sometimes these sounds fulfil biological functions and sometimes these sounds are used to communicate (e.g., to convey boredom or indicate that a faux pas has occurred). Moreover, these sounds are often used to communicate outside the context of an ongoing communicative interaction, so the contextual cues that help observers to treat linguistic stimuli as communicative are likely insufficient. These situations therefore impose inferential demands on listeners (is this vocalization communicative or just a non-communicative biological vocalization?), which parallel the ones observers face in the case of communicative action (is this movement communicative or just world-directed?). In this way, these communicative non-speech vocalizations parallel the characteristics of emblems (see main text). We therefore predict that our framework’s abstract principles apply here. In these cases, listeners may expect communicative non-speech vocalizations to reveal that they are indeed communicative. Critically, this does not imply that listeners should expect these sounds to be rare or repetitive, as these strategies were derived to reveal that a movement is not world-directed. Instead, our framework would predict that communicators should find ways to modulate the sound in a way that allows listeners to recognize that the vocalization is not a typical non-communicative vocalization.

Similarly, we also do not predict that repetition or other features would be built into complete spoken languages in order to differentiate speech from non-communicative vocalizations. While repetition is a notable form of non-arbitrariness in both signed and spoken languages, its purpose is not to signal that speech is communicative, but rather it is often used iconically to connect lexical items to their meanings [18]. However, repeated elements (i.e., reduplication) are common in baby talk (e.g., night-night and choo-choo) and these reduplicated words are easier for infants to learn [19,20]. It has been proposed that these reduplicated baby talk words “...may be more likely to be noticed in the input and stored in verbal memory than their

adult-like counterparts (e.g., *train* and *good night*), making them accessible targets for initial word learning” [21, p.1979]. Why reduplicated elements have an early learning advantage is still an open question that could relate to the identifiability of reduplicated baby talk words as discrete communicative signals.

Note that in this section, we use the term “non-speech vocalization” to refer to biological sounds such as coughing and yawning. We were not sure what the accepted term was for these sounds, so we would be happy to change the term if that’s not the case.

Small points:

3.5. pp.23-24 “Our capacity to recognize even novel movements as communicative suggests that the expectation for efficient action has broader scope than previously believed. And yet, this expectation may be more powerful than what our work captures, perhaps supporting the recognition of an even broader suite of human behaviors such as epistemic actions (like reading a book or exploring one’s surroundings) and challenging actions which are their own reward (like mastering a skill or solving a puzzle).” I didn’t understand how the expectation for efficient action might provide insight into actions like reading a book or exploring surroundings. This could do with more exposition to make the link clearer, or omitting if it is only a tenuous link.

Thank you for pointing out that this connection is unclear. Our original idea was that watching an agent stare at a sheet of paper for an extended period of time would be an irrational action, unless the sheet had a heavy amount of information in it. We thought that this type of reasoning might enable children to infer that written texts have information, and allow adults to infer how interested someone is in a document (or even how much text is in each page and the reader’s level of interest by monitoring how quickly they switch between pages). We agree that this was a tenuous link. In order to simplify our conclusion, we have omitted references to challenge-based or epistemic goals and our final paragraph now reads as follows:

Altogether, our work sheds light on the interplay between gesturers’ need to reveal when a movement is communicative, and the inferential burden this imposes on their observers. Although theories of human action understanding have classically distinguished between mechanisms that support world-directed inferences and communicative inferences, both activities are generated by minds that are motivated to fulfill their goals rationally and efficiently. Our work shows that a unified expectation for rational action enables us, as observers, to recognize what other agents are doing, whether it is acting to change the physical world, or acting to communicate. As observers, holding actors to an expectation that they are efficient and rational in both the physical and social world structures our unique ability to build nuanced models of other people’s minds from their behavior.

3.6. p.28 In the data analysis for quantification of rarity, can the authors indicate to what extent rarity is independent of distance? A small deviation from shortest distance for a short route would result in a higher rarity value than a small deviation from shortest route for a longer route,

based on how rarity is calculated. Could overall route length be a factor in affecting decisions in the studies?

Thank you for bringing this potential confound to our attention. To test for this possibility in Study 1, we computed the distance travelled for each path and correlated it against participant communicativeness judgments (combining Study 1 and its replication). The correlation between communicativeness and distance traveled was $r=0.56$, which is substantially lower than the correlation we observed between rarity and communicativeness ($r=0.85$):

We have also reconsidered all of our data in light of this concern, and we believe that Studies 2 and 3 further rule out this possibility. In Study 2, we created three versions of each path, which had different numbers of repetitions, but kept the total distance traveled constant. Here, participants rated movements with more repetitions are more likely to be communicative, despite being matched in distance to the movement with fewer repetitions. In Study 3, the distance traveled by each paired path in the bordered and unbordered conditions remains the same, but participants rated the unbordered condition paths as significantly more likely to be communicative than their bordered counterparts. Thus, Studies 2 and 3 provide even stronger evidence for rarity rather than distance as the driving factor behind communicativeness judgements. These results show that distance alone cannot explain participant communicativeness judgments.

Our revised SI for Study 1 now raises this possibility and presents these additional analyses in a section titled “Distance as a Potential Confound”.

Thank you for your time and attention and please let us know if there is any additional information that we can provide.

Sincerely,

Amanda Royka, Annie Chen, Rosie Aboody, and Julian Jara-Ettinger

REVIEWER COMMENTS

Reviewer #1 (Remarks to the Author):

I have now read the revised manuscript and the response letter in full. In brief: I am wholly impressed by the thoughtful and substantive work that went into this revision, and my concerns have been fully addressed. The substantive revisions made to the paper have successfully clarified each of the points raised. In addition, multiple literatures have been successfully brought to bear on the evidence presented, and the links to these literatures have substantially strengthened the paper, making its connections and potential for broad impact clear. I am now convinced that the evidence provides clear support for the claims of the paper, and the nature of the claims have been usefully clarified. An additional experiment, conducted to address a point I raised in my response & now included in the supplement (the relation to the account presented by Schachner & Carey, 2013), adds additional clarity, and builds toward a unified account of action understanding for actions that are not world-directed (e.g. gesture, ritual, communication, dance).

Overall, I believe this paper is now suitable for publication in its current form. This manuscript represents a valuable contribution to the literature, and I look forward to seeing it in published form.

Reviewer #3 (Remarks to the Author):

I felt the authors dealt with the reviewers' points very thoroughly. The data are convincing to me, though the Discussion suffers a little bit from having to address so many disparate points, however, it does provide an even-handed approach, and the key results and contributions of the paper are clear and seem to me substantial.

There is one issue still remaining from my previous review which needs a slight adjustment in analysis.

I was initially concerned with the issue of a potential confound of distance relating to rarity - the new analyses are welcome where distance is shown to correlate with communicativeness judgments in Study 1. Studies 2 and 3 seem to control for this effectively, so this is not a substantial issue, however, Study 1 might still point to issues about signaling of communicativeness with distance - a longer distance (greater investment, reduced efficiency?) could be potential important indicators of communicativeness as a further factor in addition to repetition and rarity. I don't think this needs a lot more discussion, but I would have liked to have seen the correlation between communicativeness and rarity when distance is partialled out. Currently, the SI (pp.10-11) describe that distance and communicativeness are correlated but less than rarity and communicativeness, and conclude from this that rarity is an additional factor - it would be useful to show that directly.

Three further minor points.

p.11 "Instead, we found no relation between rarity and difficulty of explanation (Explanation Control: $r=0.10$, $CI95\%=-0.31-0.56$; replication: $r=0.38$, $CI95\%=0.14-0.83$; Fig. 2d)". But is that significant? It is a small, but fairly consistent effect? It would be useful to include p-values in the report of the results in the main paper here to enable readers to be sure these effect sizes are not meaningful.

p.11 maybe better to say instead of "Thus, responses in Studies 1-3 likely reflect participants' belief that a communicative goal was a better explanation" this: "Thus, responses in Studies 1-3 likely primarily reflect participants' belief that a communicative goal was a better explanation"

p.7 SI: typo: "couching" should be "coughing".

Signed review: Padraic Monaghan

Reviewer #4 (Remarks to the Author):

Review of NCOMMS-21-07437A

I was asked to review only about the cross-cultural aspects of the study and will do so.

Strictly from a cross-cultural methodological perspective, the studies conducted are very weak. The authors simply develop methods and utilize stimuli developed in one culture and deploy them for use in another culture, assuming that the data obtained are valid and equivalent across cultures. Contemporary cross-cultural methodologists will quickly note the lack of attention to the concept of equivalence throughout the methods of Studies 4-7. This lack of attention to the concept of equivalence occurs in

1. The samples – the samples are clearly not equivalent in their demographics and all other characteristics, both cultural and non-cultural
2. The validity of the stimuli – the authors fail to bring to bear any evidence for the validity of the stimuli to portray accurately what is intended in the stimuli in the non-U.S. culture. There is no evidence for the equivalence in criteria (what were they, anyway) and meaning of the four versions – basic, rare, repetitive, and rare+repetitive.
3. No evidence for the equivalence of the cover stories used.
4. No evidence for the equivalence in the procedures utilized and the context in which data were obtained.

Of course, in some or even many situations, demonstrating equivalence is impossible and the non-equivalence is exactly what comprises interesting cultural differences. However, the main lack of equivalence was in the validity and meaning of the stimuli. If their meaning, or the categories used to describe them, differed between the cultures, then findings between the cultures are difficult to interpret, and conclusions are difficult to draw.

For relevant readings on cross-cultural methodology covering these issues, I suggest the authors consult van di Vijver, Poortinga, Matsumoto, Fischer and others.

RESPONSE TO REVIEWER COMMENTS

We present all reviewer comments in blue 12pt font, all of our responses in black 12pt font, and all manuscript quotes in black 10pt font with a half-inch indentation.

Reviewer #1 (Remarks to the Author):

I have now read the revised manuscript and the response letter in full. In brief: I am wholly impressed by the thoughtful and substantive work that went into this revision, and my concerns have been fully addressed. The substantive revisions made to the paper have successfully clarified each of the points raised. In addition, multiple literatures have been successfully brought to bear on the evidence presented, and the links to these literatures have substantially strengthened the paper, making its connections and potential for broad impact clear. I am now convinced that the evidence provides clear support for the claims of the paper, and the nature of the claims have been usefully clarified. An additional experiment, conducted to address a point I raised in my response & now included in the supplement (the relation to the account presented by Schachner & Carey, 2013), adds additional clarity, and builds toward a unified account of action understanding for actions that are not world-directed (e.g. gesture, ritual, communication, dance). Overall, I believe this paper is now suitable for publication in its current form. This manuscript represents a valuable contribution to the literature, and I look forward to seeing it in published form.

We are grateful for this reviewer's earlier comments. We are glad that our changes have satisfied their thoughtful questions and critiques.

Reviewer #3 (Remarks to the Author):

I felt the authors dealt with the reviewers' points very thoroughly. The data are convincing to me, though the Discussion suffers a little bit from having to address so many disparate points, however, it does provide an even-handed approach, and the key results and contributions of the paper are clear and seem to me substantial.

We are once again grateful for Dr. Monaghan's kind words and thoughtful feedback. Having taken some time away from the manuscript, we have reread the discussion and made several revisions that increase the clarity and flow,

while maintaining the same main points. We hope that the revised discussion is easier to follow, but we would gladly make further revisions if desired.

More specifically, we made three changes to our discussion. First, we have edited the writing throughout to simplify sentences and remove minor tangential points where possible. Second, we have re-organized the discussion so that it now shows a more natural flow following the structure: (1) summary, (2) limitations, (3) open questions, (4) relation to other areas of research, and (5) conclusions. Finally, to make this structure more transparent, our discussion now highlights this structure at the end of paragraph 2 (key part highlighted):

Altogether, our studies suggest that rarity and repetition support communicative inferences because they efficiently reveal non-world-directedness in a wide range of contexts. In the remainder of the discussion, we begin by considering two key limitations left unaddressed by our current work and then discuss two open questions for future investigation. Finally, we elaborate on the connections between our framework and related areas of research.

There is one issue still remaining from my previous review which needs a slight adjustment in analysis.

I was initially concerned with the issue of a potential confound of distance relating to rarity - the new analyses are welcome where distance is shown to correlate with communicativeness judgments in Study 1. Studies 2 and 3 seem to control for this effectively, so this is not a substantial issue, however, Study 1 might still point to issues about signaling of communicativeness with distance - a longer distance (greater investment, reduced efficiency?) could be potential important indicators of communicativeness as a further factor in addition to repetition and rarity. I don't think this needs a lot more discussion, but I would have liked to have seen the correlation between communicativeness and rarity when distance is partialled out. Currently, the SI (pp.10-11) describe that distance and communicativeness are correlated but less than rarity and communicativeness, and conclude from this that rarity is an additional factor - it would be useful to show that directly.

We agree that this is an informative additional analysis. To keep our analysis approach consistent throughout the paper, we have opted to use a mixed-effects model to calculate the relationship between communicativeness and rarity while controlling for distance. This approach is mathematically equivalent, because a partial correlation is the square root of the variance explained in a mixed-effects

model. However, we would be happy to switch this approach for a partial correlation if this would be preferred.

We have now run a linear mixed effects model predicting participant judgments from path distance and rarity (using the combined data from Study 1 and its replication). In this regression, we found a significant effect of rarity and a marginal effect of distance. We now include this additional analysis and discuss potential interpretations in the SI (new text highlighted below):

Testing for Effects of Distance as a Potential Confound. Study 1 began by operationalizing our notion of rarity in terms of inefficiency (although our computational model uses a richer definition in terms of non-world-directedness), defined as the ratio between the length of the observed path, and the length of the shortest possible path to reach the end point. Therefore, rarity in this study correlates with path length. This raises the possibility that participant communicative judgments were driven by overall distance travelled. To test for this possibility, we computed the distance travelled for each path and correlated it against participant communicativeness judgments (combining Study 1 and its replication). The correlation between communicativeness and distance traveled was $r = 0.56$ (Supplementary Fig. 3), which is substantially lower than the correlation we observed between rarity and communicativeness ($r = 0.85$).

Additionally, we ran linear mixed effects models for Study 1 and its replication examining the relationship between participant judgements, distance, and rarity. The model predicted participant judgments from path distance and rarity with random intercepts for path class and participant. In this regression, distance marginally predicted participant judgements ($\beta_{\text{distance}} = 0.004$, $p = 0.06$), but the beta coefficient was orders of magnitude less than the beta coefficient of rarity ($\beta_{\text{rarity}} = 2.40$, $p < 0.001$). Thus, even when partialling out the effect of distance on communicativeness judgements, rarity remains a strong predictor of communicativeness judgements. The small effect of distance may hint at a possible separate effect on communicativeness (perhaps as a proxy for effort), but more systematic manipulations focusing on distance are needed to support this hypothesis.

Disentangling the role of rarity and path length in Study 1 is difficult as these two naturally correlate. Note, however, that Studies 2 and 3 further rule out the possibility that participants responded based on path length. In Study 2, we created three versions of each path, which varied in the number of repetitions while keeping the total distance traveled constant. Here, participants rated movements with more repetitions are more likely to be communicative, despite being matched in distance to the movement with fewer repetitions. In Study 3, the distance traveled by each paired path in the bordered and unbordered conditions was held constant, but participants rated the unbordered condition paths as significantly more likely to be communicative than their bordered counterparts. Thus, Studies 2 and 3 provide even stronger evidence for rarity rather than distance as the driving factor behind communicativeness judgements. These results show that distance alone cannot explain participant communicativeness judgments.

Three further minor points.

p.11 “Instead, we found no relation between rarity and difficulty of explanation (Explanation Control: $r=0.10$, $CI95\%=-0.31-0.56$; replication: $r=0.38$, $CI95\%=0.14-0.83$; Fig. 2d)”. But is that significant? It is a small, but fairly consistent effect? It would be useful to include p-values in the report of the results in the main paper here to enable readers to be sure these effect sizes are not meaningful.

Thank you. We have added a mixed-effects regression for the Explanation Control and its replication, which shows that the relation between rarity and difficulty of explanation is not significant. We now include this in the manuscript and have adjusted the language to note it is a small effect (rather than no relation at all):

Instead, we found no significant relation between rarity and difficulty of explanation (Explanation Control: $r=0.10$, $CI95\%=-0.31-0.56$, $\beta_{rarity} = 0.21$, $p = 0.43$; replication: $r=0.38$, $CI95\%=0.14-0.83$, $\beta_{rarity} = 0.40$, $p = 0.19$; Fig. 2d), suggesting that people can easily conceive of non-communicative goals for complex movements. Thus, responses in Studies 1-3 likely primarily reflect participants’ belief that a communicative goal was a better explanation, rather than defaulting to this answer due to an inability to think of alternatives (see SI for additional analyses revealing this null effect was not due to task misunderstanding, and Studies 6-7 for additional evidence that rare and repetitive movements are actively seen as communicative).

p.11 maybe better to say instead of “Thus, responses in Studies 1-3 likely reflect participants’ belief that a communicative goal was a better explanation” this: “Thus, responses in Studies 1-3 likely primarily reflect participants’ belief that a communicative goal was a better explanation”

We have revised this sentence as suggested above.

p.7 SI: typo: “couching” should be “coughing”.

Thank you for bringing this to our attention. We have fixed this error.

Signed review: Padraic Monaghan

Reviewer #4 (Remarks to the Author):

Review of NCOMMS-21-07437A

I was asked to review only about the cross-cultural aspects of the study and will do so.

Strictly from a cross-cultural methodological perspective, the studies conducted are very weak. The authors simply develop methods and utilize stimuli developed in one culture and deploy them for use in another culture, assuming that the data obtained are valid and equivalent across cultures. Contemporary cross-cultural methodologists will quickly note the lack of attention to the concept of equivalence throughout the methods of Studies 4-7. This lack of attention to the concept of equivalence occurs in.

Thank you for reviewing our manuscript. We agree that issues of equivalence are key in designing and interpreting cross-cultural studies and have made several changes to our manuscript to highlight interpretation concerns that arise from this consideration.

We have adjusted our manuscript accordingly, highlighting caution when interpreting our cross-cultural data, and we explain these changes in response to the points below.

We also want to clarify three omissions (now included in our manuscript), which are relevant here:

1. While it is true that our initial studies (Studies 1-4) were designed in the US, all three studies run with the Tsimane' (Studies 5-7) were specifically designed for use with that community and then subsequently replicated in the US. This was done in consultation with local Tsimane'-Spanish bilinguals who grew up in Tsimane' communities and helped ensure the clarity of the task. In the Methods section, we now clarify as follows:

All studies with Tsimane' participants were conducted with a local interpreter. All task paradigms and instructions were discussed in advance with the interpreters to ensure clarity and fluidity. Specifically, all Tsimane' studies were designed with feedback from Tsimane' interpreters to ensure that the cover stories, critical task elements (e.g., the idea of communicating through movements that might look different than conventional gestures used by the participants), and test questions, were clear and that their phrasing did not have any unforeseen implications arising from translation. Participants were allowed to participate in no more than one study.

2. Our tasks were intentionally designed to minimize the amount of language needed to explain its logic, helping us reduce concerns about lack of equivalence in translation. Each task had a single short cover story and used simple response measures that minimized the possibility of misinterpretation. While this does not render a task equivalent, it does differentiate our task from language-heavy surveys, in which stimuli may lack item equivalence (such as the surveys used as case studies in the work the reviewer helpfully pointed us towards in their last point).

3. We agree that direct cross-cultural comparisons without equivalence can be problematic. Our original manuscript was not sufficiently careful about this and, at several points, our writing invited a comparison. However, it is also worth noting that none of our main results involve cross-cultural comparisons, and they were instead limited to within-culture comparisons. That is, our results do not hinge on comparing data from one population to another. Instead, we analyze each population separately, and focus on how their pattern of responses changes as a function of different visual stimuli (in this case, body movements). One exception to this are two regressions that were originally included in the supplemental text (but not discussed in main text) for completeness. We have since removed these regressions from the manuscript.

It is our understanding that issues with equivalence become a central concern in three situations. The first is when a task is designed in one culture and simply imported into another, which can lead to issues of construct validity. The second, is when the task uses language-based surveys which can be difficult to translate, opening the space for errors or unintended implications of language. The third is when a research study aims to interpret differences in task performance as implying cross-cultural differences in a cognitive construct.

In this project, we did not import a task into the Tsimane', but re-designed it from the ground up; we did not use language-heavy surveys, but minimally-linguistic tasks designed in close consultation with Tsimane' locals; and we do not make any claims about cross-cultural differences (beyond noting a hypothesis that differences in effect sizes may reflect comfort with experimental tasks).

That being said, we take seriously the issues raised by Reviewer 4, and we have edited our manuscript in several ways to clearly mark the limitations of the cross-cultural dimension of our study. We review the edits in our responses below.

1. The samples – the samples are clearly not equivalent in their demographics and all other characteristics, both cultural and non-cultural

The reviewer is correct: We did not try to match samples along demographic features because we did not predict a priori that age, gender, or schooling should affect people's judgements about novel communicative actions.

Fortunately, we collected enough data with the Tsimane' to sufficiently power a regression that can isolate the potential contribution of any of these demographic features. We ran a mixed effects model predicting whether or not the participant's response aligned with our hypothesis based on age, gender, and schooling with intercepts for study and participant (for Studies 5-7). We find that age and gender do not have a significant effect on the probability that participants responded in line with our hypotheses ($\beta_{\text{age}} = 0.002$, $p = 0.592$; $\beta_{\text{gender}} = -0.17$, $p = 0.204$), but years of schooling does have a small significant effect improving the probability that the participant will respond in line with our hypotheses ($\beta_{\text{schooling}} = 0.04$, $p = 0.008$). We believe this effect of education is consistent with related findings in the Tsimane' showing that education yields an improvement on experimental tasks, possibly due to increased comfort on experimental tasks (Gibson et al., 2017). We now present this regression in our supplemental materials, and highlight that our results should be taken only as initial findings due to concerns about equivalence:

A priori, we did not expect that age, gender, or schooling should affect people's judgements about novel communicative actions. However, in response to a reviewer request, here we analyze the effect of demographic variables on the performance of Tsimane' participants. Specifically, we conducted a mixed effects model predicting whether or not the participant's response aligned with our hypothesis based on age, gender, and schooling with intercepts for study and participant (for Studies 5-7). We find that age and gender do not have a significant effect on the probability that participants responded in line with our hypotheses ($\beta_{\text{age}} = 0.002$, $p = 0.592$; $\beta_{\text{gender}} = -0.17$, $p = 0.204$), but years of schooling does have a small significant effect improving the probability that the participant will respond in line with our hypotheses ($\beta_{\text{schooling}} = 0.04$, $p = 0.008$). We believe this effect of education is consistent with related findings in the Tsimane' showing that education yields an improvement on experimental tasks, possibly due to increased comfort in experimental tasks (Gibson et al., 2017). However, it is likely that years of schooling in the Tsimane' correlate with other life experiences. Therefore, further work is needed to delve into this issue and as such our cross-cultural findings should be treated as only preliminary evidence of the effect of rarity and repetition on communicativeness judgements outside of a US context.

2. The validity of the stimuli – the authors fail to bring to bear any evidence for the validity of the stimuli to portray accurately what is intended in the stimuli in the non-U.S. culture. There is no evidence for the equivalence in criteria (what were they, anyway) and meaning of the four versions – basic, rare, repetitive, and rare+repetitive.

We apologize for the confusion. The four categories are meant to refer to three objective features of the stimuli, not subjective psychological constructs. In the video stimuli, rarity means the movement did not include physical contact between body parts, and repetition means the movement was repeated three times.

The reviewer is correct that our results implicitly assume that Tsimane' participants may see the repetitive movements as repetitive, and the rare movements as uncommon. We do not know if this is the case. Although Tsimane' participants show the same pattern of responses as US participants, we agree that it is possible that this was purely coincidental, with the Tsimane' conceptualizing the body movements in a radically different way.

To clarify this point, we have carefully edited the manuscript throughout so that the results from the Tsimane' are explained in terms of their response to rare and repetitive videos (i.e., using rare and repetitive as the video classification), rather than rare and repetitive movements (i.e., which can be taken as referring to a psychological construct).

3. No evidence for the equivalence of the cover stories used.

4. No evidence for the equivalence in the procedures utilized and the context in which data were obtained.

It is worth noting that all the studies run with Tsimane' and US participants (Studies 5, 6, and 7) were designed for the Tsimane' and then adapted to be run in the US, rather than the other way around.

From this standpoint, Study 4 can be understood as the US-centric study and Study 5 as the Tsimane'-centric equivalent study. This is why Study 6 tests the same question as Study 5, but does so using a different cover story, different warm-up procedure (using examples of locally known gestures to give examples of what we meant when explaining the idea that body movements can be used to communicate), and different dependent variable (using forced-choices rather than numerical scales). Because we also expected some readers to want to see

data under closely-matched experimental set-ups, we then adapted the paradigms designed for the Tsimane' and tested them in the US.

Although Study 5 was designed in close consultation with local bilingual Tsimane' community members, and ethnography experts, we agree that we do not have conclusive evidence for equivalence (as the reviewer notes below, demonstrating equivalence is often impossible). It was in part because of this concern that we decided to run two additional studies, Study 6 and 7, which target the same phenomena using different methodologies. We therefore now present our studies as a first step, and explicitly note their limitations. When introducing the cross-cultural dimension of our work we write:

To what extent are these intuitions unique to people in the US? According to our proposal, communicative action is detected by relying on basic action-understanding mechanisms that emerge early in infancy (3-6, 8), and these inferences might therefore show cross-cultural stability. As a first step in this question, we evaluated these intuitions with the Tsimane'—a farming-foraging group living in the Bolivian Amazon. The Tsimane' have historically lived in relative isolation and, consequently, differ from Western participants in several areas including the timing of the acquisition of number words and concepts (35-36), aesthetic preferences in sound (37), and color vocabulary (38). Studies 5-7 were designed in consultation with our interpreters and local experts, and then replicated in the US.

Of course, in some or even many situations, demonstrating equivalence is impossible and the non-equivalence is exactly what comprises interesting cultural differences. However, the main lack of equivalence was in the validity and meaning of the stimuli. If their meaning, or the categories used to describe them, differed between the cultures, then findings between the cultures are difficult to interpret, and conclusions are difficult to draw.

We agree with the reviewer, and thank them for raising this important point. We have edited the manuscript throughout accordingly. After discussing possibilities for the qualitative differences of Study 5, we write:

However, it is also possible that the difference in effect sizes emerged because participants from different cultures interpreted the task in different ways (known as a lack of equivalence; 40-44), and our results should be interpreted with caution. As such, these results represent a first step in establishing that Tsimane' and San Borjan participants are also more likely to endorse rare and repetitive movements as potentially communicative.

We have also removed all language in the main text implying any direct cross-cultural comparison. We agree that direct cross-cultural comparisons without

equivalence could be problematic, and as such we limit ourselves to within culture comparisons and simply point out that they show the same patterns.

For relevant readings on cross-cultural methodology covering these issues, I suggest the authors consult van di Vijver, Poortinga, Matsumoto, Fischer and others.

Thank you for this suggestion. We have read and discussed work from these authors and find ourselves in agreement with their arguments about the nuances of cross-cultural studies, which helped shape the comments we provide above and edits to our manuscript. Additionally, we now cite several of these papers in the manuscript when stating the limitations of our cross-cultural work.

Please let us know if there is any further information we can provide.

Sincerely,

Amanda Royka, Annie Chen, Rosie Aboody, Tomas Huanca, and Julian Jara-Ettinger

References cited in this response:

Gibson, E., Jara-Ettinger, J., Levy, R., & Piantadosi, S. (2017). The use of a computer display exaggerates the connection between education and approximate number ability in remote populations. *Open Mind*, 2(1), 37-46.

Huanca, T. (2008). Tsimane' oral tradition, landscape, and identity in tropical forest. SEPHIS.

REVIEWERS' COMMENTS

Reviewer #3 (Remarks to the Author):

This revision addresses all my previous concerns, and I think it is an excellent contribution to the field. I don't have expertise on the cross-cultural issues that the new reviewer 4 provided, so cannot comment on the effectiveness of the response there, but for the the rest I'm happy to recommend acceptance.

Reviewer #4 (Remarks to the Author):

I find the authors replies and the associated changes in the ms to meet all my concerns to the degree that they can be reasonably addressed given the nature of the tasks and samples. Thanks for being attentive to those issues.

REVIEWERS' COMMENTS

Reviewer #3 (Remarks to the Author):

This revision addresses all my previous concerns, and I think it is an excellent contribution to the field. I don't have expertise on the cross-cultural issues that the new reviewer 4 provided, so cannot comment on the effectiveness of the response there, but for the the rest I'm happy to recommend acceptance.

Reviewer #4 (Remarks to the Author):

I find the authors replies and the associated changes in the ms to meet all my concerns to the degree that they can be reasonably addressed given the nature of the tasks and samples. Thanks for being attentive to those issues.

We are grateful for the reviewers' feedback and are glad that our revisions have sufficiently addressed their concerns.